# FLEXIBLE HETEROSCEDASTIC COUNT REGRESSION WITH DEEP DOUBLE POISSON NETWORKS

## ABSTRACT

Neural networks that can produce accurate, input-conditional uncertainty representations are critical for real-world applications. Recent progress on heteroscedastic *continuous* regression has shown great promise for calibrated uncertainty quantification on complex tasks, like image regression. However, when these methods are applied to *discrete* regression tasks, such as crowd counting, ratings prediction, or inventory estimation, they tend to produce predictive distributions with numerous pathologies. Moreover, discrete models based on the Generalized Linear Model (GLM) framework either cannot process complex input or are not fully heterosedastic. To address these issues we propose the Deep Double Poisson Network (DDPN). In contrast to networks trained to minimize Gaussian negative log likelihood (NLL), discrete network parameterizations (i.e., Poisson, Negative binomial), and GLMs, DDPN can produce discrete predictive distributions of arbitrary flexibility. Additionally, we propose a technique to tune the prioritization of mean fit and probabilistic calibration during training. We show DDPN 1) vastly outperforms existing discrete models; 2) meets or exceeds the accuracy and flexibility of networks trained with Gaussian NLL; 3) produces proper predictive distributions over discrete counts; and 4) exhibits superior out-of-distribution detection. DDPN can easily be applied to a variety of count regression datasets including tabular, image, point cloud, and text data.

## 1 INTRODUCTION

The pursuit of neural networks capable of learning accurate and reliable uncertainty representations has gained significant traction in recent years (Lakshminarayanan et al., 2017; Kendall & Gal, 2017; Gawlikowski et al., 2023; Dheur & Taieb, 2023). Input-dependent uncertainty is useful for detecting out-of-distribution data (Amini et al., 2020; Liu et al., 2020; Kang et al., 2023), active learning (Settles, 2009; Ziatdinov, 2024), reinforcement learning (Yu et al., 2020; Jenkins et al., 2022), and real-world decision-making under uncertainty (Abdar et al., 2021). While uncertainty quantification applied to regression on continuous outputs is well-studied, training neural networks to make probabilistic predictions over discrete counts has traditionally received less attention, despite multiple relevant applications. In recent years, neural networks have been trained to predict the size of crowds (Zhang et al., 2016; Lian et al., 2019; Zhang & Chan, 2020; Zou et al., 2019; Luo et al., 2020; Lin & Chan, 2023), the number of cars in a parking lot (Hsieh et al., 2017), traffic flow (Lv et al., 2014; Liu et al., 2021; Li et al., 2020), agricultural yields (You et al., 2017), inventory of product on shelves (Jenkins et al., 2023), and bacteria in microscopic images (Marsden et al., 2018). In this paper, we are interested in training neural networks to output a flexible, calibrated, and properly specified predictive distribution over discrete counts (Figure 1).

A common approach to uncertainty representation in complex regression tasks has been to apply the generalized linear model (GLM) framework, but to replace the linear predictor with a neural network. The network is then trained to output the mean and variance of a Gaussian distribution, $\left[\hat{\mu}_i, \hat{\sigma}_i^2\right]^T = \mathbf{f}_\Theta(\mathbf{x}_i)$ (Nix & Weigend, 1994), while minimizing Gaussian negative log likelihood (NLL) loss via gradient-based optimization. This form of input-conditional predictive variance is known as *heteroscedastic* regression. Recent work has improved the performance of heteroscedastic regression by mediating the influence of $\hat{\sigma}^2$ on the gradient of the mean, which can cause instability during training, miscalibrated predictive variance, or a poor mean fit (Immer et al., 2024; Seitzer et al., 2022; Stirn et al., 2023).

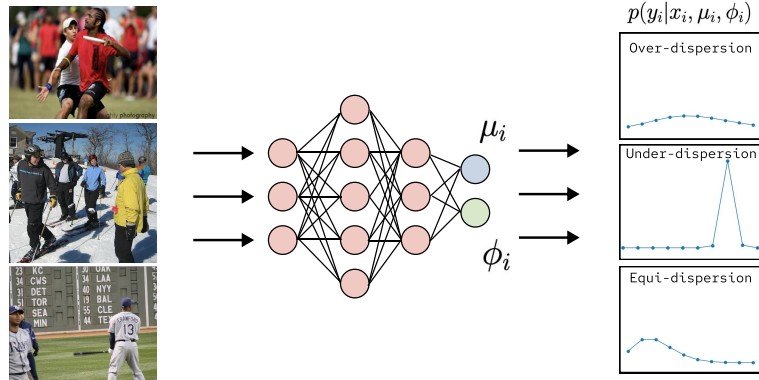

Figure 1: An overview of the Deep Double Poisson Network (DDPN) and discrete heteroscedastic regression problem. A deep neural network processes complex data as input (i.e. image, text or point cloud) and outputs the parameters of a *discrete* probability distribution over an integer prediction range. The mean, $\mu_i$, and inverse dispersion, $\phi_i$, vary for each input, $x_i$ and allow for over-, under-, and equi-dispersion.

However, when each of these methods is applied to count regression, the model is trained to output an input-dependent probability *density* function, $p(y|\mathbf{f}_\Theta(\mathbf{x}));\ y \in \mathbb{R}$, over a *discrete* output space, i.e. $y \in \mathbb{Z}_{\geq 0}$. Applying a continuous density function to a discrete domain creates three critical pathologies. First, the continuous predictive distribution will assign non-zero probability mass to infeasible real values that fall in between valid integers. Second, the predictive intervals are unbounded and can assign non-zero probability to negative values when the predicted mean is small. Third, the boundaries of the predictive intervals (i.e., high density interval or 95% credible interval) are likely to fall between two valid integers, diminishing their interpretability and utility. To overcome these limitations, we desire a properly specified probability *mass* function, conditional on the input features: $p(y|\mathbf{f}_\Theta(\mathbf{x}));\ y \in \mathbb{Z}_{\geq 0}$.

Discrete regression has historically been treated similarly to the Gaussian case. For example, previous work trains a network to predict the $\lambda$ parameter of a Poisson distribution and minimize its NLL (Fallah et al., 2009). However, the Poisson parameterization of the neural network suffers from the *equi-dispersion* assumption: predictive mean and variance of the Poisson distribution are the same ($\hat{\lambda} = \hat{\mu} = \hat{\sigma}^2$). Therefore, the model is not flexible enough to produce separate input-dependent mean and variance predictions. Another common alternative is to train the network to minimize Negative Binomial (NB) NLL (Xie, 2022). The Negative Binomial breaks *equi-dispersion* by introducing another parameter to the PMF. This helps disentangle the mean and variance, but suffers from the *over-dispersion* assumption: $\hat{\sigma}^2 \geq \hat{\mu}$. Consequently, this model is not flexible enough to assign uncertainty less than its mean prediction for a given input. Meanwhile, discrete GLMs fit without a neural network feature extractor lack representational capacity to process complex input and are also not fully heteroscedastic (Efron, 1986; Murphy, 2023).

**Our Contributions** To address these issues, we introduce the Deep Double Poisson Network (DDPN), a novel discrete neural regression model (See Figure 1). In contrast to Gaussian-based heteroscedastic regressors, DDPN is a neural network trained to output the parameters of the Double Poisson Distribution (Efron, 1986), which represents a highly flexible, discrete predictive distribution, $p(y|\mathbf{f}_\Theta(\mathbf{x}));\ y \in \mathbb{Z}_{\geq 0}$. DDPN is fully heteroscedastic such that the predicted mean and dispersion are independent, conditioned on the input. Additionally, we demonstrate that DDPN is subject to similar dynamics between mean and dispersion during training as Gaussian-based techniques (Immer et al., 2024; Seitzer et al., 2022; Stirn et al., 2023), and propose a $\beta$ modification to the NLL to temper this relationship and achieve 'tunable mean fit' (Figure 3). Compared to existing discrete regression models, DDPN is flexible enough to handle over-, under- and equi-dispersion, making it a superior choice to the Poisson and Negative Binomial deep networks for discrete predictive uncertainty quantification. Our experiments show that DDPN can learn accurate and reliable uncertainty representations on both tabular and complex data (image, point cloud, and text). DDPN matches or exceeds the performance and calibration of Gaussian-based alternatives and offers superior out-of-distribution detection compared with existing techniques.

| Method | Discrete Predictive | Complex Data | Fully Heteroscedastic | Tunable Mean Fit |
|---|---|---|---|---|
| Discrete GLMs | ✓ | ✗ | ✗ | ✗ |
| Gaussian DNN | ✗ | ✓ | ✓ | ✗ |
| Gaussian $\beta$-NLL | ✗ | ✓ | ✓ | ✓ |
| Poisson/NB DNN | ✓ | ✓ | ✗ | ✗ |
| DDPN (ours) | ✓ | ✓ | ✓ | ✓ |

Table 1: Summary of contributions and existing work.

## 2 Modeling Predictive Uncertainty with Neural Networks

Predictive uncertainty can be decomposed into two types: *epistemic* (uncertainty of the model weights) and *aleatoric* uncertainty (observation noise) (Kendall & Gal, 2017; Der Kiureghian & Ditlevsen, 2009).

### 2.1 Epistemic Uncertainty

Epistemic uncertainty refers to uncertainty due to model misspecification. Modern neural networks tend to be significantly underspecified by the data, which introduces a high degree of uncertainty (Wilson & Izmailov, 2020). In general, this type of uncertainty can be reduced through additional data acquisition. A variety of techniques have been proposed to explicitly represent epistemic uncertainty including Bayesian inference (Wilson & Izmailov, 2020; Chen et al., 2014; Hoffman et al., 2014), variational inference (Graves, 2011), and Laplace approximation (Daxberger et al., 2021). Recently, deep ensembles have emerged as a simple and popular alternative (Lakshminarayanan et al., 2017; D'Angelo & Fortuin, 2021). Other work connects Bayesian inference and ensembles by arguing the latter can viewed as a Bayesian model average where the posterior is sampled at multiple local modes (Fort et al., 2019; Wilson & Izmailov, 2020). This approach has a number of attractive properties: 1) it generally improves predictive performance (Dietterich, 2000); 2) it can model more complex predictive distributions; and 3) it effectively represents uncertainty over learned weights, which leads to better probabilistic calibration.

### 2.2 Heteroscedastic Regression for Aleatoric Uncertainty

Aleatoric uncertainty quantifies observation noise and generally cannot be reduced with more data (Der Kiureghian & Ditlevsen, 2009; Kendall & Gal, 2017). In practice, this uncertainty can be introduced by low resolution sensors, blurry images, or the intrinsic noise of a signal. Aleatoric noise is commonly modeled in machine learning by fitting the parameters of a distribution over the output, rather than a point prediction. Uncertainty is often represented by a dispersion parameter, $\sigma$, that is learned from the training data. When dispersion varies for each input, $\sigma_i$, we get a heteroscedastic model. Below, we detail how aleatoric uncertainty is modeled in both the GLM and deep learning literature.

#### 2.2.1 Generalized Linear Models

Under this paradigm, observation noise is modeled by specifying a conditional distribution, $p(y_i|\eta_i, \sigma)$, where $p$ is a member of the exponential family, $\eta_i = \boldsymbol{w}^T \boldsymbol{x}_i$ represents the natural parameter of $p$, and $\sigma$ is the dispersion term (McCullagh, 2019; Murphy, 2023). A link function, $l(\cdot)$, is selected to specify a mapping between the natural parameter and the mean such that $l(\mu_i) = \eta_i = \boldsymbol{w}^T \boldsymbol{x}_i$. The model is then fit by minimizing NLL. Many common models can be viewed under this general framework, including logistic regression, Poisson regression, and binomial regression (Fahrmeir et al., 2013). It was in this setting that the Double Poisson distribution was first introduced. However, initial models with this distribution were strictly linear, and constrained the dispersion term with an explicit dependence on the mean. Specifically, given parameter vectors $\boldsymbol{\alpha}$ and $\boldsymbol{\beta} = [\beta_0, \beta_1, \beta_2]^T$, Efron (1986) assumes $\log(\hat{\mu}_i) = \hat{\eta}_i = \boldsymbol{\alpha}^T \boldsymbol{x}_i$ and $\hat{\sigma}_i = \frac{M}{1+e^{-(\beta_0 + \beta_1 \hat{\mu}_i + \beta_2 \hat{\mu}_i^2)}}$.

This approach has two key limitations: 1) the predicted dispersion, $\hat{\sigma}_i$, does not directly depend on the input $\boldsymbol{x}_i$, and is instead a function of the predicted mean, $\hat{\mu}_i$); 2) the hyperparameter $M$ introduces an upper bound on the dispersion, which in turn curtails the feasible range of confidence values. In practice, the authors set $M = 1.25$, which hardly allows for under-dispersion ($\sigma > 1$). Both of

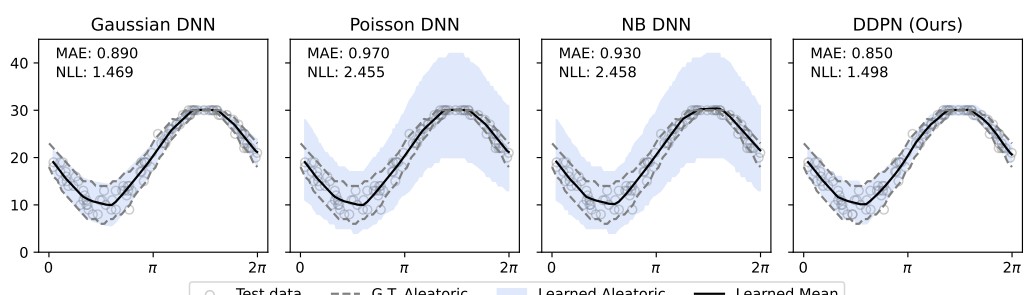

Figure 2: Simulation experiment with a known data-generating process featuring heteroscedastic variance over discrete outputs. Here we model varying levels of dispersion, with severe under-dispersion on the high values of $y$ and increased spread on the low values of $y$. We show the ground truth aleatoric uncertainty interval and the test data points. We visualize the mean fit and "learned" aleatoric uncertainty (centered 95% credible interval of the predictive distribution) of each of 4 probabilistic neural networks on the test split of the dataset, along with the mean absolute error (MAE) and NLL. All models adequately fit the mean. However, only the `Gaussian DNN` and `DDPN` correctly recover the heteroscedastic pattern in all regions. The `Poisson DNN` and `NB DNN` lack sufficient flexibility to capture under-dispersion.

these measures significantly limit the family of distribution functions the model can learn. Follow-up studies all assume a constant dispersion term, applying even stronger limits on flexibility (Toledo et al., 2022; Zhu, 2012; Zou et al., 2013). In contrast to these, our proposed approach drastically expands the family of functions that can be modeled. DDPN learns a non-linear mapping that can be trained on complex data and can fully disentangle the mean and dispersion, allowing for pure heteroscedastic regression. We also introduce a tunable hyperparameter that allows for custom prioritization between mean fit and overall likelihood calibration.

### 2.2.2 GENERALIZED REGRESSION WITH DEEP LEARNING

GLMs are limited in their predictive power and cannot fit complex data. To address this, similar theoretical principles have been applied to specify deep neural networks, which are much more flexible than GLMs and map to a larger number of data modalities (Fallah et al., 2009; Xie, 2022; Qi et al., 2020; Barron, 2019; Fan et al., 2019). These works adjust the natural parameter mapping as follows: let $\mathbf{z} = \mathbf{g}_{\boldsymbol{\Theta}_{1:L-1}}(\mathbf{x}_i)$ denote the features extracted by the first $L-1$ layers of a neural network. Both the mean and dispersion are outputs of the network: $l(\mu_i) = \eta_i = \boldsymbol{w}_\eta^T \mathbf{z}(\mathbf{x}_i)$, $\sigma_i = \boldsymbol{w}_\sigma^T \mathbf{z}$. One specific example of this approach is detailed in both Nix & Weigend (1994) and Kendall & Gal (2017), where the network is trained to output the mean and log variance of a Gaussian, $[\hat{\mu}_i, \log \hat{\sigma}_i]^T = \mathbf{f}_{\boldsymbol{\Theta}}(\mathbf{x}_i)$ with the objective of minimizing Gaussian NLL. Recent work has identified issues with this training strategy due to the the influence of $\hat{\sigma}$ on the mean, $\hat{\mu}$. Immer et al. (2024) reparameterize the neural network to output the natural parameters of the Gaussian distribution. Seitzer et al. (2022) propose a modified loss function and introduce a hyperparameter, $\beta \in [0, 1]$, which tempers the impact of $\hat{\sigma}^2$ on the gradient of the mean. Stirn et al. (2023) re-scale the gradient of $\hat{\mu}$ and modify the architecture of the underlying network to include separate sub-networks for $\hat{\mu}$ and $\hat{\sigma}^2$, along with the stop gradient operation to prevent the gradient of $\hat{\sigma}^2$ from impacting the $\hat{\mu}(\mathbf{x})$ sub-network.

For count regression, one can specify a neural network that outputs the parameters of a discrete distribution. For example, Fallah et al. (2009) train a neural network to predict the mean and variance parameter, $\lambda$, of a Poisson distribution, while Xie (2022) applies this idea to the Negative Binomial distribution. As discussed previously, these approaches suffer from the equi- and over-dispersion assumptions. In contrast, DDPN produces a fully heteroscedastic, discrete predictive distribution, and offers tunable mean fit through a likelihood $\beta$ modification.

## 3 DEEP DOUBLE POISSON NETWORKS (DDPN)

In this section, we introduce the Deep Double Poisson Network (DDPN), which is a neural network that outputs the parameters of the Double Poisson distribution (Efron, 1986; Toledo et al., 2022). The

main idea of DDPN is to flexibly and accurately model an input-conditional predictive distribution over the space of discrete counts (See Figure 1). We propose 1) a fully heteroscedastic parameterization that disentangles predicted mean and dispersion conditioned on the input, 2) a novel loss function based on the Double Poisson likelihood (Equation 2), and 3) the introduction of a hyperparameter, $\beta$, to the loss function that offers tunable prioritization between fitting the natural likelihood and mean accuracy.

We assume access to a dataset, $\mathcal{D}$, with $N$ training examples $\{\mathbf{x}_i, y_i\}_{i=1}^N$, where each $y_i \in \mathbb{Z}_{\geq 0}$ is drawn from some unknown nonnegative discrete distribution $p(y_i|\mathbf{x}_i)$. Let $\mathcal{X}$ denote the space of all possible inputs $\mathbf{x}$, let $\mathcal{P}$ denote the space of all possible distributions over $\mathbb{Z}_{\geq 0}$, and let $\boldsymbol{\psi} \in \mathbb{R}^d$ denote a vector of parameters identifying a specific $p \in \mathcal{P}$. We wish to model $\mathcal{P}$ with a neural network $\mathbf{f}_{\boldsymbol{\Theta}} : \mathcal{X} \to \mathcal{P}$ with learnable weights $\boldsymbol{\Theta}$. In practice, we model $\mathbf{f}_{\boldsymbol{\Theta}} : \mathcal{X} \to \boldsymbol{\psi} \in \mathbb{R}^d$. Given such a network, we obtain a predictive distribution, $\hat{p}(y|\mathbf{f}_{\boldsymbol{\Theta}}(\mathbf{x}))$, for any input $\mathbf{x}$.

In particular, suppose that we restrict our output space to $\mathcal{P}_{DP} \subset \mathcal{P}$, the family of Double Poisson distributions over $y$. Any distribution $p \in \mathcal{P}_{DP}$ is uniquely parameterized by $\boldsymbol{\psi} = [\mu, \phi]^T \succ \mathbf{0}$, for mean, $\mu$, and inverse dispersion, $\phi$. The distribution function, $p : \mathbb{Z}_{\geq 0} \to [0, 1]$, is defined as follows (where $c$ is a normalizing constant):

$$p(y|\mu, \phi) = \frac{\phi^{\frac{1}{2}} e^{-\phi\mu}}{c(\mu, \phi)} \left( \frac{e^{-y} y^y}{y!} \right) \left( \frac{e\mu}{y} \right)^{\phi y}, \; c(\mu, \phi) \approx 1 + \frac{1 - \phi}{12\mu\phi} \left( 1 + \frac{1}{\mu\phi} \right) \tag{1}$$

Let $Z$ denote a random variable with a Double Poisson distribution function (Equation 1). Then we say $Z \sim \mathrm{DP}(\mu, \phi)$, with $\mathbb{E}[Z] \approx \mu$ and $\mathrm{Var}[Z] \approx \frac{\mu}{\phi}$ (Efron, 1986). We specify a model [1], $[\log \hat{\mu}_i, \log \hat{\phi}_i]^T = \mathbf{f}_{\boldsymbol{\Theta}}(\mathbf{x}_i)$, with the following structure: let $\mathbf{z}_i = \mathbf{g}_{\boldsymbol{\Theta}_{1:L-1}}(\mathbf{x}_i)$, be the $d$-dimensional hidden representation of the input $\mathbf{x}_i$ produced by the previous $L - 1$ layers. We then apply two separate linear layers to this hidden representation to obtain our distribution parameters: $\log(\hat{\mu}_i) = \boldsymbol{w}_\mu^T \mathbf{z}_i$ and $\log(\hat{\phi}_i) = \boldsymbol{w}_\phi^T \mathbf{z}_i$. In contrast to previous work described in Section 2.2.1, this parameterization allows for fully disentangled mean and dispersion predictions, conditioned on the hidden representation of the input. Additionally, this removes the constraint, $M$, on the dispersion and allows for arbitrary sharpness of the predictive distribution.

## 3.1 DDPN Objective

To learn the weights we minimize the following objective based on Double Poisson NLL:

$$\mathcal{L}_{DDPN}(y_i, \hat{\mu}_i, \hat{\phi}_i) = \frac{1}{N} \sum_{i=1}^N \left( -\frac{1}{2} \log \hat{\phi}_i + \hat{\phi}_i \hat{\mu}_i - \hat{\phi}_i y_i (1 + \log \hat{\mu}_i - \log y_i) \right) \tag{2}$$

During training, we minimize $\mathcal{L}_{DDPN}$ iteratively via stochastic gradient descent (or common variants). We provide a full derivation of Equation 2 in Appendix A.3.

## 3.2 $\beta$-DDPN: NLL Loss Modifications

As first noted in Seitzer et al. (2022), when training a heteroscedastic regressor with Gaussian likelihood, the ability of a neural network to fit the mean can be harmed by the presence of the predicted variance term in the partial derivative of the mean. We observe that this same phenomenon exists with DDPN. We have the following partial derivatives with respect to $\hat{\mu}_i$ and $\hat{\phi}_i$:

$$\frac{\partial \mathcal{L}_{DDPN}}{\partial \hat{\mu}_i} = \hat{\phi}_i \left( 1 - \frac{y_i}{\hat{\mu}_i} \right), \quad \frac{\partial \mathcal{L}_{DDPN}}{\partial \hat{\phi}_i} = -\frac{1}{2\hat{\phi}_i} + \hat{\mu}_i - y_i(1 + \log \hat{\mu}_i - \log y_i) \tag{3}$$

---

[1]For both $\hat{\mu}$ and $\hat{\phi}$ we apply the log "link" function to ensure positivity and numerical stability. We simply exponentiate whenever $\hat{\mu}_i$ or $\hat{\phi}_i$ are needed (i.e., to evaluate the density function in Equation 1)

Notice that if $\hat{\phi}_i$ is sufficiently small (corresponding to large variance), it can completely zero out $\frac{\partial \mathscr{L}_{DDPN}}{\partial \hat{\mu}_i}$ regardless of the current value of $\hat{\mu}_i$. Thus, during training, a neural network can converge to (and get "stuck" in) suboptimal solutions wherein poor mean fit is explained away via large uncertainty values. To remedy this behavior, we propose a modified loss function, the $\beta$-DDPN:

$$\mathscr{L}_{\beta-DDPN}(y_i, \hat{\mu}_i, \hat{\phi}_i) = \frac{1}{N} \sum_{i=1}^{N} \left\lfloor \hat{\phi}_i^{-\beta} \right\rfloor \left( -\frac{1}{2} \log \hat{\phi}_i + \hat{\phi}_i \hat{\mu}_i - \hat{\phi}_i y_i (1 + \log \hat{\mu}_i - \log y_i) \right) \quad (4)$$

where $\lfloor \cdot \rfloor$ denotes the *stop-gradient* operation. With this modification we can effectively temper the effect of large variance on mean fit. We now have the following partial derivatives:

$$\frac{\partial \mathscr{L}_{\beta-DDPN}}{\partial \hat{\mu}_i} = \left( \hat{\phi}_i^{1-\beta} \right) \left( 1 - \frac{y_i}{\hat{\mu}_i} \right), \quad \frac{\partial \mathscr{L}_{\beta-DDPN}}{\partial \hat{\phi}_i} = -\frac{1}{2\hat{\phi}_i^{1+\beta}} + \hat{\mu}_i - y_i (1 + \log \hat{\mu}_i - \log y_i)$$

$$(5)$$

The Double Poisson $\beta$-NLL is parameterized by $\beta \in [0, 1]$, where $\beta = 0$ recovers the original Double Poisson NLL and $\beta = 1$ corresponds to fitting the mean, $\mu$, with no respect to $\phi$ (while still performing normal weight updates to fit the value of $\phi$). Thus, we can consider the value of $\beta$ as providing a smooth interpolation between the natural DDPN likelihood and a more mean-focused loss (Figure 3). For an empirical demonstration of the impact of $\beta$ on DDPN, see Figure 5.

### 3.3 DDPN Ensembles

The formulation of DDPN described above applies to neural networks with a single forward pass. As noted in Section 2, multiple independently trained neural networks can be combined to improve mean fit and distributional calibration by modeling epistemic uncertainty. Thus, we propose a technique for constructing an ensemble of DDPNs to further enhance the quality of the predictive distribution. Following Lakshminarayanan et al. (2017) and Fort et al. (2019), we train $M$ different DDPNs on the *same* dataset and only vary the random initialization point. This produces $M$ different solutions $\{\Theta_m\}_{m=1}^{M}$ yielding $M$ distinct predictive distributions for any given input, $\{p(y_i|\mathbf{f}_{\Theta_m}(\mathbf{x}_i))\}_{m=1}^{M}$. For our ensemble prediction, we form a uniform mixture of each distribution: $p(y_i|\mathbf{x}_i) = \frac{1}{M} \sum_{m=1}^{M} p(y_i|\mathbf{f}_{\Theta_m}(\mathbf{x}_i))$. In Appendix A.5 we provide well-known equations for recovering the mean and variance of this mixture distribution (Marron & Wand, 1992).

Figure 3: Effect of the proposed $\beta$ modification. The partial derivative of the likelihood *w.r.t* the mean, $\frac{\partial \mathscr{L}}{\partial \mu_i}$, naturally depends on $\phi_i$, which can cause poor mean fit to be explained away via large uncertainty values, harming accuracy. Increasing $\beta$ reduces this dependency.

## 4 Experiments

We evaluate DDPN across a variety of count regression tasks based on tabular, image, point cloud, and text data. Each dataset has been divided using a 70-10-20 train/val/test split with a fixed random seed (results are reported on the test split). We compare a number of baselines, including a Poisson Generalized Linear Model (GLM), a Negative Binomial GLM, a Double Poisson GLM (Efron, 1986; Toledo et al., 2022; Zhu, 2012; Zou et al., 2013), a Gaussian Deep Neural Network (DNN) (Nix & Weigend, 1994), a Poisson DNN (Fallah et al., 2009), Negative Binomial DNN (Xie, 2022), the "faithful" DNN regressor presented in Stirn et al. (2023), the naturally parameterized Gaussian regressor from Immer et al. (2024), and the reparameterized network (with $\beta = 0.5$, as recommended) from Seitzer et al. (2022). Additionally, we show the impact of the $\beta$-DDPN modification (with subscripts indicating the exact value of $\beta$) presented in Section 3.2. We refer to these as "single forward pass" methods. We also ensemble our method and compare to ensembles of Gaussian, Poisson, and Negative Binomial DNNs to demonstrate the impact of modeling both

aleatoric and epistemic uncertainty. Gaussian ensembles are formed using the technique introduced in Lakshminarayanan et al. (2017), while Poisson and Negative Binomial ensembles follow the same prediction strategy outlined in Section 3.3. All experiments are implemented in PyTorch (Paszke et al., 2017). Choices related to network architecture, hardware and hyperparameter selection are reported in Appendix B. Source code is freely available online[2].

Each regression method is evaluated in terms of two criteria. First, Mean Absolute Error (MAE) measures the predictive accuracy and mean fit; lower values imply higher accuracy. Second, Negative Log Likelihood (NLL) measures the calibration, or quality, of the predictive distribution (Candela et al., 2005); lower values imply greater agreement between the predictive distribution $p$ and the observed label $y_i$. We choose to omit the commonly-used ECE (Kuleshov et al., 2018) as a measure of calibration due to its recently identified shortcomings when evaluating discrete probability distributions (Young & Jenkins, 2024). To facilitate comparison between NLL obtained from continuous and discrete models, we use the continuity correction to convert Gaussian densities into probabilities. Given a predicted Gaussian CDF $\hat{F}_i$ for some input-output pair $(x_i, y_i)$, we take $P(Y = y_i | \hat{F}_i) \approx \hat{F}_i(y_i + \frac{1}{2}) - \hat{F}_i(y_i - \frac{1}{2})$. We then compute NLL as the average of $-\log P(Y = y_i | \hat{F}_i)$ across the evaluation set. For each technique, we train and evaluate 5 models and report the empirical mean and standard deviation (in parentheses). To form ensembles, these same 5 models are combined.

### 4.1 Simulation Experiments

To clearly illustrate the flexibility of the DDPN in modeling count data, we simulate a dataset that exhibits varying levels of dispersion. The exact data generating process is described in Appendix B.1. We train a small multi-layer perceptron (MLP) to output the parameters of a Gaussian, Poisson, Negative Binomial, or Double Poisson distribution using the appropriate NLL loss. The resultant models' predictive distributions over the test split of the synthetic dataset are visualized in Figure 2. MAE and NLL are both reported in each panel of the figure.

DDPN clearly meets or exceeds the flexibility and accuracy of the Gaussian while maintaining a proper distribution over discrete counts. It achieves slightly better mean fit (lower MAE) and roughly equivalent calibration (NLL). Conversely, the Poisson and Negative Binomial DNNs lack the capacity to recover the heteroscedastic variance pattern of the data. For another simulated demonstration of DDPN's flexibility, see Appendix A.2, where we show DDPN can recover the ground-truth conditional dependencies in the data even when explicitly misspecified.

### 4.2 Tabular Datasets

We perform two experiments on tabular datasets, one with high frequency counts, and one with low frequency counts. The `Bikes` dataset (Fanaee-T & Gama, 2014) describes the number of hourly bike rentals between the years 2011 and 2012 in the Capital bikeshare system. The features are the corresponding weather and seasonal information. The 25th, 50th, and 75th percentiles of the labels, $y_i$, are (40, 142, 281), indicating high frequency events. The `Collision` dataset (for Transport, 2022) is formed from the casualties, collisions, and vehicles tables in the United Kingdom's 2022 Road Safety data. In this task, the goal is to predict the number of casualties in a collision, given features about the accident (i.e., drivers, vehicles, location, etc.). The labels are severely right-skewed, ranging from 1 to 16 with a mean of 1.278 and a median of 1. For each dataset, we train an MLP to output the parameters of each benchmarked distribution. See Table 2 for results.

In `Bikes` we observe DDPN surpasses state-of-art heteroscedastic Gaussian regression baselines in terms of mean fit and approaches the performance of the Poisson DNN. We note that Poisson likely performs well because the provided features are not sufficient for concentrated predictions and the data are naturally equi- to over-dispersed. On the other hand, both DDPN and $\beta_{1.0}$-DDPN outperform all methods in terms of probabilistic fit (NLL). In `Collision`, we see that $\beta_{0.5}$-DDPN and $\beta_{1.0}$-DDPN top the baselines in terms of mean fit while maintaining competitive NLL with the DP GLM. DDPN also performs well on these two dimensions and is close to Seitzer and NB DNN in terms of mean fit. In both cases, modeling epistemic uncertainty via ensembling provides significant improvements in mean fit and calibration, with DDPN outperforming alternatives.

---

[2]https://anonymous.4open.science/r/ddpn-651F/README.md

|  |  | Bikes | | Collision | |
|---|---|---|---|---|---|
|  |  | MAE ($\downarrow$) | NLL ($\downarrow$) | MAE ($\downarrow$) | NLL ($\downarrow$) |
| Single Forward Pass | Poisson GLM | 110.07 (2.59) | 9.81 (0.02) | 0.394 (0.00) | 1.186 (0.01) |
|  | NB GLM | 190.03 (0.00) | 10.83 (0.09) | 0.322 (0.02) | 1.120 (0.01) |
|  | DP GLM | 164.43 (8.87) | 8.71 (0.79) | 0.271 (0.00) | **0.675** (0.00) |
|  | Gaussian DNN | 38.70 (2.65) | 5.00 (0.04) | 0.305 (0.00) | 0.772 (0.10) |
|  | Poisson DNN | **27.76** (0.34) | 5.81 (0.04) | 0.316 (0.01) | 1.181 (0.00) |
|  | NB DNN | 32.33 (6.71) | 4.72 (0.08) | 0.277 (0.00) | 1.183 (0.01) |
|  | Stirn et al. (2023) | 28.54 (0.40) | 5.07 (0.06) | 0.302 (0.00) | 1.005 (0.00) |
|  | Seitzer et al. (2022) | 38.64 (0.80) | 5.01 (0.05) | 0.274 (0.00) | 0.722 (0.00) |
|  | Immer et al. (2024) | 35.30 (0.74) | 5.03 (0.04) | 0.304 (0.00) | 0.723 (0.00) |
|  | DDPN (ours) | 28.18 (0.34) | **4.67** (0.01) | 0.280 (0.00) | 0.719 (0.01) |
|  | $\beta_{0.5}$-DDPN (ours) | 30.36 (1.06) | 4.73 (0.03) | **0.269** (0.00) | 0.710 (0.01) |
|  | $\beta_{1.0}$-DDPN (ours) | 28.93 (0.80) | 4.70 (0.01) | **0.269** (0.00) | 0.707 (0.01) |
| Deep Ensembles | Gaussian DNN | 34.40 | 4.87 | 0.282 | 0.756 |
|  | Poisson DNN | 26.01 | 5.15 | 0.278 | 1.178 |
|  | NB DNN | 28.00 | 4.62 | **0.270** | 1.179 |
|  | DDPN (ours) | **25.96** | **4.57** | 0.271 | **0.610** |
|  | $\beta_{0.5}$-DDPN (ours) | 27.30 | 4.65 | **0.270** | 0.703 |
|  | $\beta_{1.0}$-DDPN (ours) | 26.37 | 4.60 | **0.270** | 0.697 |

Table 2: Results on tabular datasets: We report the Mean Absolute Error (MAE) and Negative Log Likelihood (NLL) for each method. We denote the best performer in **bold** and the second-best performer with an underline.

Although the linear models we measure perform adequately on `Collision`, they struggle with the more complex feature interactions in `Bikes`, thus failing to model the true data distribution. This supports our commentary on the built-in rigidity of GLMs in Section 2.2.1. Overall, our results suggest that DDPN is effective in the tabular regime for both high and low-frequency counts.

## 4.3 COMPLEX DATASETS

We introduce an image regression task on the `person` class of MS-COCO (Lin et al., 2014), which we call `COCO-People`. In this dataset, the task is to predict the number of people in each image. We also define an inventory counting task (Jenkins et al., 2023), where the goal is to predict the number of objects on a retail shelf from an input point cloud (see Figure 21 in the Appendix for an example). Finally, we predict discrete user ratings from the "Patio, Lawn, and Garden" split of a collection of Amazon reviews (Ni et al., 2019). The objective in this task is to predict the discrete review value (1-5 stars) from an input text sequence, which historically has been addressed with Gaussian NLL (Mnih & Salakhutdinov, 2007; Koren et al., 2009). For `COCO-People`, each model was trained with a small MLP on top of the pooled output from a ViT backbone (initialized from the `vit-base-patch16-224-in21k` checkpoint (Wu et al., 2020; Deng et al., 2009)). For the `Inventory` dataset, each model was fitted with a variant of CountNet3D (Jenkins et al., 2023) that was modified to output the parameters of a distribution instead of regressing the mean directly. All text regression models were constructed as a small MLP on top of the `[CLS]` token in the output layer of a DistilBert backbone (starting from the `distilbert-base-cased` checkpoint) (Sanh et al., 2019). See Table 3 for results.

In `COCO-People` we see strong performance in terms of both mean fit (MAE) and calibration (NLL), with either DDPN or $\beta_{1.0}$-DDPN leading all methods. As expected, DDPN outperforms benchmarks in terms of calibration, while $\beta_{1.0}$-DDPN yields the best mean performance. We show example predictions from the `COCO-People` test set in Appendix C.1. In `Inventory`, DDPN and $\beta_{1.0}$-DDPN achieve the best mean fit, with the slight edge in NLL going to DDPN. `Reviews` sees $\beta_{0.5}$-DDPN and $\beta_{1.0}$-DDPN score favorably in terms of mean fit, essentially matching the predictive performance of Stirn. Immer yields the best results in terms of probabilistic fit, with DDPN close behind.

One note of interest is that although the $\beta = 0.5$ setting appears to yield slightly worse individual DDPNs on `COCO-People` and `Inventory`, these models make for an excellent predictive ensemble, achieving top scores across the board for `Inventory`, the best MAE for `Reviews`, and second

| | | COCO-People | | Inventory | | Reviews | |
|---|---|---|---|---|---|---|---|
| | | MAE ($\downarrow$) | NLL ($\downarrow$) | MAE ($\downarrow$) | NLL ($\downarrow$) | MAE ($\downarrow$) | NLL ($\downarrow$) |
| *Single Forward Pass* | Gaussian DNN | 2.010 (0.03) | 2.308 (0.02) | 0.904 (0.01) | 1.559 (0.01) | 0.326 (0.01) | 0.834 (0.09) |
| | Poisson DNN | 2.013 (0.14) | 2.393 (0.08) | 0.960 (0.02) | 1.763 (0.00) | 0.609 (0.04) | 1.705 (0.00) |
| | NB DNN | 2.082 (0.30) | 2.284 (0.04) | 0.965 (0.01) | 1.801 (0.03) | 0.746 (0.09) | 1.711 (0.00) |
| | Stirn et al. (2023) | 2.045 (0.20) | 2.490 (0.08) | 0.927 (0.03) | 1.651 (0.02) | **0.301** (0.00) | 0.878 (0.02) |
| | Seitzer et al. (2022) | 2.279 (0.14) | 2.450 (0.05) | 0.907 (0.02) | 1.610 (0.03) | 0.307 (0.00) | 0.940 (0.24) |
| | Immer et al. (2024) | 2.129 (0.26) | 2.359 (0.09) | 0.925 (0.02) | 1.587 (0.02) | 0.310 (0.00) | **0.728** (0.01) |
| | DDPN (ours) | 2.148 (0.23) | **2.251** (0.06) | **0.900** (0.01) | **1.555** (0.01) | 0.311 (0.00) | 0.800 (0.01) |
| | $\beta_{0.5}$-DDPN (ours) | 2.300 (0.69) | 2.395 (0.14) | 0.902 (0.00) | 1.625 (0.05) | 0.302 (0.00) | 1.531 (0.53) |
| | $\beta_{1.0}$-DDPN (ours) | **1.962** (0.35) | 2.517 (0.15) | **0.900** (0.01) | 1.560 (0.02) | 0.302 (0.00) | 1.027 (0.15) |
| *Deep Ensembles* | Gaussian DNN | 1.941 | 2.195 | 0.873 | 1.511 | 0.306 | **0.726** |
| | Poisson DNN | 1.875 | 2.141 | 0.924 | 1.754 | 0.600 | 1.702 |
| | NB DNN | 1.849 | 2.073 | 0.902 | 1.790 | 0.750 | 1.707 |
| | DDPN (ours) | 1.904 | 1.962 | 0.861 | 1.500 | 0.295 | 0.729 |
| | $\beta_{0.5}$-DDPN (ours) | 1.824 | 1.916 | **0.839** | **1.469** | **0.274** | 0.825 |
| | $\beta_{1.0}$-DDPN (ours) | **1.701** | **1.891** | 0.851 | 1.486 | 0.281 | 0.753 |

Table 3: Results on complex datasets: COCO-People (image), Inventory (point cloud), and Amazon Reviews (language). We denote the best performer in **bold** and the second-best performer with an underline.

place in MAE and NLL on `COCO-People`. One potential explanation is that $\beta = 0.5$ encourages a greater diversity of learned models, which lends itself favorably to capturing epistemic uncertainty. In general, we see superior results when ensembling DDPN variants as compared to other models.

### 4.4 OUT-OF-DISTRIBUTION BEHAVIOR

In this section, we compare the out-of-distribution (OOD) behavior of DDPNs to existing methods. To assess OOD behavior, for each model that has been trained on `Reviews`, we feed it verses from the King James Version of the Holy Bible, and compute the entropy (Shannon, 1948) of each of the resultant predictive distributions; we call these OOD entropy values. We do the same with the test split of `Reviews`, and call them in-distribution (ID) entropy values. We then compare the empirical distributions of these entropy values (Amini et al., 2020) by performing a one-sided permutation test (Good, 2013) on the difference of means. This procedure outputs a test statistic, $\Delta = \bar{x}_{OOD} - \bar{x}_{ID}$, and a p-value (for more details see Appendix B.5). Higher entropy indicates higher uncertainty in a model's predictive distributions. Thus, we expect that the models most able to distinguish between ID / OOD will have the larger $\Delta$ since their mean entropy should be higher for OOD inputs than ID inputs.

The results of our experiment are displayed in Figure 4. With statistical significance, DDPN shows the greatest ability of all benchmarked regression models to differentiate between ID and OOD inputs, as demonstrated by the largest $\bar{\Delta}$ (the average $\Delta$ across trials). Existing count regression techniques (NB DNN, Poisson DNN) fail to exhibit any separation between predictive entropy on ID and OOD data. We note that only half of Gaussian regression approaches benchmarked (Immer et al., 2024; Seitzer et al., 2022) achieve a significant gap between ID and OOD entropies. For a similar analysis showing the supremacy of DDPN *ensemble* methods in terms of OOD behavior, see Figure 8 in the Appendix. We provide a case study of OOD detection in Appendix C.2. In particular, Figure 12 highlights the effective OOD behavior of DDPN.

In Section 3.2 we discussed the motivation for $\beta$-DDPN as a mechanism to prioritize mean accuracy over probabilistic calibration. Empirically, this hypothesis is generally supported by our experiments. The $\beta$ modification that is used to enhance mean fit appears to hurt a model's recognition of OOD. From all experiments, our general conclusion is the virtue of $\beta$-DDPN is highly accurate mean prediction, while the advantage of standard DDPN is reliable calibration and effective OOD detection.

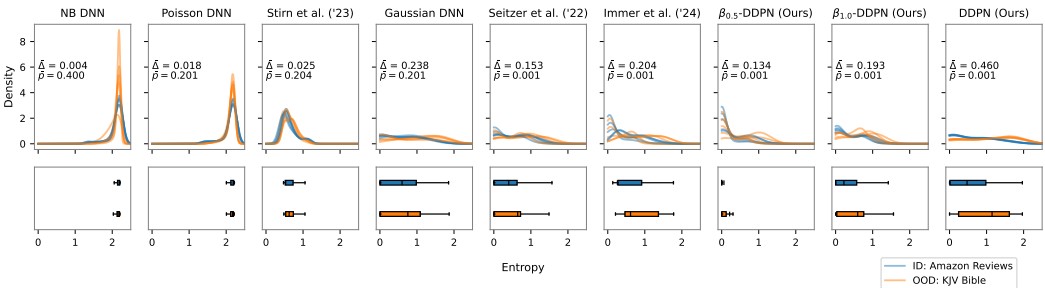

Figure 4: In-distribution (ID) vs. out-of-distribution (OOD) behavior for regression models trained on `Amazon Reviews`. We train each method five times, and plot the KDE-smoothed empirical distributions of entropy values obtained from the ID (`Amazon Reviews`) and OOD (`KJV Bible`) datasets. Additionally, we provide a box plot with an IQR of aggregated entropy values. We perform a two-sample permutation test with the difference-of-means statistic ($\Delta$) and display, on the corresponding KDE plot, the average statistic ($\bar{\Delta}$) across all models, along with the average p-value ($\bar{p}$). A larger $\bar{\Delta}$ is desirable, as it corresponds to a greater amount of entropy on OOD than ID inputs. Our DDPN model shows the greatest ability to distinguish between ID and OOD inputs.

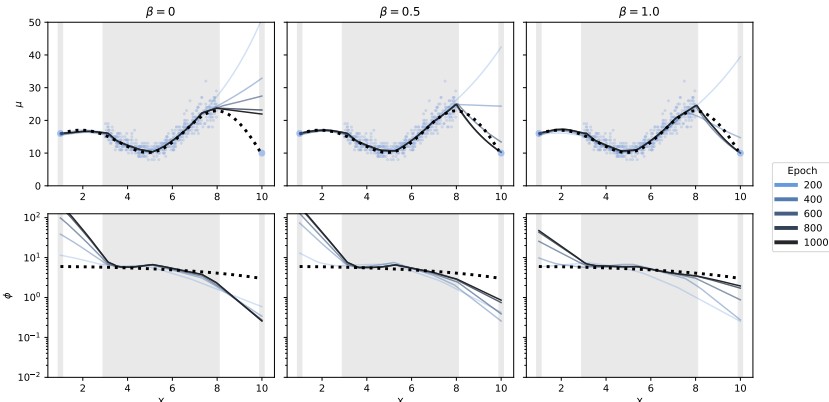

Figure 5: Demonstration of the effect of $\beta$ on the convergence of a DDPN during training, inspired by Fig. 2 of Stirn et al. (2023). Data was drawn from $Y|X \sim \mathrm{DP}(\lceil X\sin(X) + 15 \rceil, 6 - 0.03X^2)$, where $X \sim \mathrm{Uniform}[3, 8]$, and was then concatenated with isolated points $(1, \lceil \sin(1) + 15 \rceil)$ and $(10, \lceil 10\sin(10) + 15 \rceil)$. Dotted lines indicate g.t. values of $\mu$ and $\phi$ respectively, while solid lines show the model's learned distribution. Shaded regions illustrate training data coverage. With pure Double Poisson NLL, poor mean fit on the rightmost isolated point is "explained away" via high uncertainty (low values of $\phi$), leading to subpar convergence to the true data-generating distribution. Increasing the value of $\beta$ changes training priorities and allows the network to adequately model the mean without exploding uncertainty estimates. Higher values of $\beta$ lead to faster convergence to the mean; when $\beta = 0.5$, the mean is fit by epoch 800, but when $\beta = 1.0$, the mean is fit by epoch 600.

## 5 CONCLUSION

Overall, we conclude that DDPNs are well-suited for complicated count regression tasks. Our main findings are that DDPNs 1) vastly outperform existing deep learning methods with discrete predictive distributions; 2) match or exceed the performance of state-of-the-art heteroscedastic regression techniques; 3) address pathologies with Gaussian-based heteroscedastic regressors applied to discrete counts; and 4) provide superior out-of-distribution detection, compared to existing methods. Moreover, DDPNs are general and can be applied to a variety of tasks and data modalities.

## 6    ETHICS STATEMENT

We have reviewed the ICLR Code of Ethics and affirm our commitment to upholding it. We are not aware of any violations of this code associated with our research.

## 7    REPRODUCIBILITY STATEMENT

We have made a sizeable effort to ensure reproducibility of our experimental results. These include providing extensive architectural and computational details, hyperparameter specifications, and optimizer configurations, along with an exact statement of the objective functions used to train our DDPN models. We also provide a link to an anonymized repository in Footnote 2 with the source code we used to obtain our results.

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

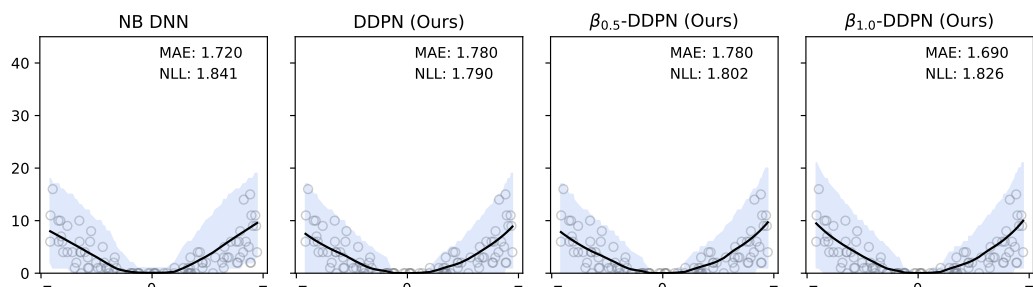

Figure 6: Results of training a DDPN when the data-generating process is Negative Binomial. The dark line depicts the mean of the model's predictive distribution, while shaded regions indicate the model's learned aleatoric uncertainty, similar to Figure 2. DDPN (along with its $\beta$ variants) is able to recover the ground-truth distribution better than a NB DNN, even though it is technically "misspecified".

## A  DEEP DOUBLE POISSON NETWORKS (DDPNs)

### A.1  LIMITATIONS

DDPNs are general, easy to implement, and can be applied to a variety of datasets. However, some limitations do exist. One limitation that might arise is on count regression problems of very high frequency (i.e., on the order of thousands or millions). In this paper, we don't study the behavior of DDPN relative to existing benchmarks on high counts. In this scenario, it is possible that the choice of a Gaussian as the predictive distribution may offer a good approximation, even though the regression targets are discrete.

We also note that the general approximations $\mathbb{E}[Z] \approx \mu$ and $\mathrm{Var}[Z] \approx \frac{\mu}{\phi}$ for some $Z \sim \mathrm{DP}(\mu, \phi)$ we employ in this work have not been extensively studied. It is possible that there are edge cases where these estimates diverge from the true moments of the distribution.

One difficulty that can sometimes arise when training a DDPN is poor convergence of the model weights. In preliminary experiments for this research, we had trouble obtaining consistently high-performing solutions with the SGD (Kiefer & Wolfowitz, 1952) and Adam (Kingma & Ba, 2014) optimizers, thus AdamW (Loshchilov & Hutter, 2017) was used instead. Future researchers using the DDPN technique should be wary of this behavior.

In this paper, we performed a single out-of-distribution (OOD) experiment on `Amazon Reviews`. This experiment provided encouraging evidence of the efficacy of DDPN for OOD detection. However, the conclusions drawn from this experiment may be somewhat limited in scope since the experiment was performed on a single dataset and task. Future work should seek to build off of these results to more fully explore the OOD properties of DDPN on other count regression tasks.

### A.2  MISSPECIFICATION RECOVERY

Here we study how well DDPN can recover the true data generating function, even when the data are drawn from a non-double poisson distribution. We simulate a dataset as follows: Let $X \sim \mathrm{Uniform}[-3, 3]$ and $Y|X \sim \mathrm{NegBinom}(X^2, 0.5)$. We train a NegBinom DNN (Xie, 2022), a DDPN, a $\beta_{0.5}$-DDPN, and a $\beta_{1.0}$-DDPN each with the same MLP backbone (see Section B.1 of the Appendix for specific architecture details). A depiction of the learned distributions can be seen in Figure 6, with the MAE and NLL indicated in each panel. These results suggest that even when the data-generating process is not strictly Double Poisson, the flexibility of DDPN allows it to recover the ground-truth distribution anyway.

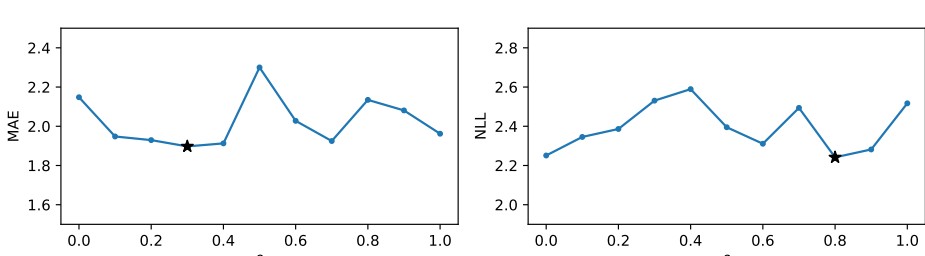

Figure 7: Results of a grid search for DDPN models with differing values of $\beta$ trained on COCO People. The best-performing value of each metric is starred.

### A.3 DERIVING THE DDPN OBJECTIVE

This loss function is obtained by first noting that

$$
\max_{\boldsymbol{\Theta}} \left[ \frac{1}{N} \sum_{i=1}^{N} p(y_i | \mathbf{f}_{\boldsymbol{\Theta}}(\mathbf{x}_i)) \right] = \max_{\boldsymbol{\Theta}, \boldsymbol{\mu}, \boldsymbol{\phi}} \left[ \frac{1}{N} \sum_{i=1}^{N} p(y_i | \mu_i, \phi_i) \right]
$$

$$
= \min_{\boldsymbol{\Theta}, \boldsymbol{\mu}, \boldsymbol{\phi}} \left[ -\frac{1}{N} \sum_{i=1}^{N} \log(p(y_i | \mu_i, \phi_i)) \right]
$$

$$
= \min_{\boldsymbol{\Theta}, \boldsymbol{\mu}, \boldsymbol{\phi}} \left[ -\frac{1}{N} \sum_{i=1}^{N} \log \left( \phi_i^{\frac{1}{2}} e^{-\phi_i \mu_i} \left( \frac{e^{-y_i} y_i^{y_i}}{y_i!} \right) \left( \frac{e \mu_i}{y_i} \right)^{\phi_i y_i} \right) \right]
$$

$$
= \min_{\boldsymbol{\Theta}, \boldsymbol{\mu}, \boldsymbol{\phi}} \left[ -\frac{1}{N} \sum_{i=1}^{N} \log \left( \phi_i^{\frac{1}{2}} e^{-\phi_i \mu_i} \left( \frac{e \mu_i}{y_i} \right)^{\phi_i y_i} \right) \right]
$$

$$
= \min_{\boldsymbol{\Theta}, \boldsymbol{\mu}, \boldsymbol{\phi}} \left[ -\frac{1}{N} \sum_{i=1}^{N} \left( \frac{1}{2} \log \phi_i - \phi_i \mu_i + \phi_i y_i (1 + \log \mu_i - \log y_i) \right) \right]
$$

Thus,

$$
\mathcal{L}_{DDPN}(y_i, \mu_i, \phi_i) = \frac{1}{N} \sum_{i=1}^{N} \left( -\frac{1}{2} \log \phi_i + \phi_i \mu_i - \phi_i y_i (1 + \log \mu_i - \log y_i) \right) \tag{6}
$$

### A.4 $\beta$ GRID SEARCH ON COCO-PEOPLE

In addition to the intuition-building experiment we provide for the $\beta$-DDPN (see Figure 5), we also run a grid search on COCO-People, varying the value of $\beta$ along a mesh of values between 0 and 1. Results of this grid search can be viewed in Figure 7.

### A.5 DDPN ENSEMBLES

In Section 3.3 we describe how the ensembled predictive distribution is a uniform mixture of the $M$ members of the ensemble:

$$
p(y_i | \mathbf{x}_i) = \frac{1}{M} \sum_{m=1}^{M} p(y_i | \mathbf{f}_{\boldsymbol{\Theta}_{\mathbf{m}}}(\mathbf{x}_i)) \tag{7}
$$

Letting $\mu_m = \mathbb{E}[y_i|\mathbf{f}_{\boldsymbol{\Theta}_\mathbf{m}}(\mathbf{x}_i)]$ and $\sigma_m^2 = \mathrm{Var}[y_i|\mathbf{f}_{\boldsymbol{\Theta}_\mathbf{m}}(\mathbf{x}_i)]$, we can get the mean and variance of the predictive distribution as follows:

$$\mathbb{E}[y_i|\mathbf{x}_i] = \frac{1}{M}\sum_{m=1}^{M}\mu_m, \ \mathrm{Var}[y_i|\mathbf{x}_i] = \sum_{m=1}^{M}\frac{\sigma_m^2 + \mu_m^2}{M} - \left(\sum_{m=1}^{M}\frac{\mu_m}{M}\right)^2 \tag{8}$$

We note that this same technique can be applied to form an ensemble from any collection of neural networks outputting a discrete distribution, regardless of the specific parametric form (Marron & Wand, 1992).

# B  DETAILED DESCRIPTION OF EXPERIMENTS

In all experiments, instead of using the final set of weights achieved during training with a particular technique, we selected the weights associated with the best mean absolute error (MAE) on a held-out validation set. This can be viewed as a form of early stopping, since models were observed to eventually overfit to the training data on almost every dataset we tested.

We note that when a point prediction was required, such as for computing the MAE of a model, we took the mode of the posterior predictive distribution instead of the mean. When the mode was not an integer (e.g. in the Gaussian case), we rounded to the nearest integer.

The ReLU (Fukushima, 1969) activation was exclusively used for all MLPs. No dropout or batch normalization was applied.

## B.1  SIMULATION EXPERIMENT

This dataset is generated with the following procedure: First, we sample $x$ from a uniform distribution, $x \sim \texttt{Uniform}(0, 2\pi)$. Next, we draw an initial proposal for $y$ from a conflation (Hill, 2011) of five identical Poissons, each with rate parameterized by $\lambda(x) = 10\sin(x) + 10$. We scale $y$ by $-1$ and shift it by $+30$ to force high dispersion at low counts and under-dispersion at high counts while maintaining nonnegativity.

Each MLP (with layers of width `[128, 128, 128, 64]`) was trained for 200 epochs on the CPU of a 2021 MacBook Pro with a batch size of 32 using the AdamW optimizer (Loshchilov & Hutter, 2017). The initial learning rate was set to $10^{-3}$ and annealed to 0 with a cosine schedule (Loshchilov & Hutter, 2016), and weight decay was set to $10^{-5}$.

## B.2  TABULAR DATASETS

### B.2.1  BIKES

In this experiment, each regression head was placed on top of an MLP with layers of width `[128, 128, 128, 64]`. Models were trained for 100 epochs on the CPU of a 2021 MacBook Pro with the AdamW optimizer, using a batch size of 128. The initial learning rate was $10^{-3}$, decayed to 0 following a cosine schedule. Weight decay was set to $10^{-5}$. For continuous features such as `temperature`, model inputs were standardized to have a mean of 0 and a standard deviation of 1. The `season`, `mnth`, and `hr` columns were transformed using a trigonometric encoding procedure.

Due to the higher counts in this dataset, and to facilitate a fairer comparison, for the `Gaussian DNN`, `Stirn`, and `Seitzer` techniques, we reconfigured the model to output $[\log\hat{\mu}_i, \log\hat{\sigma}_i^2]^T$ instead of $[\hat{\mu}_i, \log\hat{\sigma}_i^2]^T$. We observed a great performance boost with this adjustment.

We used the `Bikes` dataset under the Creative Commons Attribution 4.0 International (CCBY 4.0) license. The source URL is `https://archive.ics.uci.edu/dataset/275/bike+sharing+dataset`.

### B.2.2  COLLISIONS

We formed the `Collisions` dataset by joining the "Casualties", "Collisions", and "Vehicles" tables on the `accident_reference` column. Feature engineering included merging all associated data from a specific collision into a single row (by creating columns for each feature of each vehicle involved in the collision, for example) and one-hot encoding all categorical variables. The MLP used for feature extraction had layer widths of `[1630, 512, 256, 256, 128, 128, 128, 64]`. Models were trained on a 2021 MacBook Pro CPU for 100 epochs with a batch size of 32. The AdamW optimizer was used, with an initial learning rate of $10^{-5}$ and a cosine decay to 0.

The `Collisions` dataset is published by the United Kingdom's Department for Transport, and we used it under the Open Government Licence. The URL where this data is hosted is `https://www.data.gov.uk/dataset/cb7ae6f0-4be6-4935-9277-47e5ce24a11f/road-safety-data`.

### B.3  VISION DATASETS

### B.3.1  COCO-PEOPLE

All networks were trained for 30 epochs (updating all weights, including the ViT backbone) using the AdamW optimizer with an initial learning rate of $10^{-3}$ and weight decay of $10^{-5}$. The learning rate was decayed to 0 with a cosine schedule. The regression head on top of the ViT backbone was a two-layer MLP with layer widths `[384, 256]`. Models were trained in a distributed fashion across 4 Nvidia L4 Tensor Core GPUs on a Google Cloud Platform (GCP) VM instance, with an effective batch size of 256. Images were normalized with the ImageNet (Deng et al., 2009) pixel means and standard deviations and augmented during training with the `AutoAugment` transformation (Cubuk et al., 2018). Training was done with BFloat 16 Mixed Precision.

The `COCO` dataset from which we formed the `COCO-People` subset is distributed via the CCBY 4.0 license. It can be accessed at `https://cocodataset.org/#home`.

### B.3.2  INVENTORY

Networks were trained with the AdamW optimizer for 50 epochs with an initial learning rate of $10^{-3}$ and weight decay of $10^{-5}$. Cosine annealing was used to decay the learning rate to 0. An effective batch size of 16 was used, split across an internal cluster of 4 NVIDIA GeForce RTX 2080 Ti GPUs.

The `Inventory` dataset was made available to us via an industry collaboration and is not publicly accessible.

### B.4  TEXT DATASET

### B.4.1  AMAZON REVIEWS

All networks were trained for 10 epochs across 8 Nvidia L4 Tensor Core GPUs (on a GCP VM instance) with an effective batch size of 2048. The AdamW optimizer was used for training, with an initial learning rate of $10^{-4}$ (annealed to 0 with a cosine schedule) and weight decay of $10^{-5}$. Training was done with BFloat 16 Mixed Precision. Both the feature extractor, DistilBERT (Sanh et al., 2019), and the MLP regression head (with layer widths `[384, 256]`) were updated during training.

`Amazon Reviews` is publicly available (with a citation, which we provide in the body of the paper) at `https://cseweb.ucsd.edu/~jmcauley/datasets/amazon_v2/`. The "Patio, Lawn, and Garden" subset we employ in this work is accessible at `https://datarepo.eng.ucsd.edu/mcauley_group/data/amazon_v2/categoryFilesSmall/Patio_Lawn_and_Garden.csv`.

### B.5  OUT-OF-DISTRIBUTION BEHAVIOR

We run a one-sided, two-sample permutation test (Good, 2013) using the difference of means as our test statistic. Given samples $S_{ID}$ and $S_{OOD}$ with respective means $\bar{x}_{ID}$ and $\bar{x}_{OOD}$, we

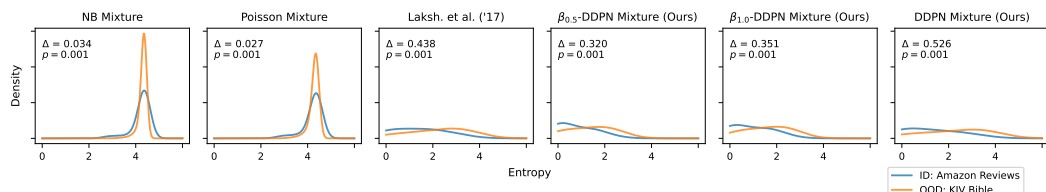

Figure 8: In-distribution (ID) vs. out-of-distribution (OOD) behavior for ensembles of regression models trained on `Amazon Reviews`. We plot the KDE-smoothed distributions of entropy values obtained from the ID (`Amazon Reviews`) and OOD (`KJV Bible`) datasets (see Section 4.4 for more details). We also perform a two-sample permutation test with the difference-of-means statistic ($\Delta$) and display the statistic, along with the p-value from the test, on the corresponding plot for each ensemble model. Just like in the individual case, ensembles of DDPN models exhibit the largest gap in entropy between ID and OOD data.

| | Inventory | | Reviews | |
|---|---|---|---|---|
| MAE ($\downarrow$) | NLL ($\downarrow$) | | MAE ($\downarrow$) | NLL ($\downarrow$) |
| 1.013 (0.02) | 1.591 (0.03) | | 0.293 (0.00) | 0.680 (0.08) |

Table 4: Multi-Class NN results on `Inventory` and `Amazon Reviews`. Compare with Table 3 in the main body of the paper.

define $\Delta = \bar{x}_{OOD} - \bar{x}_{ID}$. We then take $n = 1500$ permutations of $S_{ID}$ and $S_{OOD}$ and compute $\Delta^{(i)} = \bar{x}^{(i)}_{OOD} - \bar{x}^{(i)}_{ID}$ for each permutation $i \in \{1, 2, ..., n\}$. We take $p = \frac{|\{i \mid \Delta^{(i)} > \Delta\}|}{n}$ to be the proportion of permutations yielding a greater difference of means than $\Delta$. In a formal sense, if we define the null hypothesis $H_0 : \Delta \leq 0$ and the alternative hypothesis $H_1 : \Delta > 0$, we may treat $p$ as an estimate of $P(S_{ID}, S_{OOD}|H_0)$. Higher entropy indicates higher uncertainty / expected chaos in a model's predictive distributions. Thus, we expect that the models most able to distinguish between ID / OOD will have the highest $\Delta$ (since their mean entropy should be higher on OOD than on ID).

### B.5.1 MODELING DISCRETE COUNTS WITH A MULTI-CLASS NETWORK

In certain special cases of count regression where the targets are assumed to live on a bounded subset of $\mathbb{Z}_{\geq 0}$, it is possible to model the data via a multi-class neural network (trained with cross entropy) as opposed to an unbounded discrete probability distribution. Two of the complex datasets we benchmark in this paper can be seen as falling into this category: `Inventory` (since we expect a finite number of products to be on a given shelf) and `Reviews` (since ratings live on a fixed scale from 1 to 5). We provide metrics for a multi-class NN on these datasets in Table 4. The results are somewhat nuanced. on `Inventory`, it appears that treating a finite, discrete response as a count regression problem has clear advantages, as all models benchmarked in the main body of the paper (see Table 3) achieve superior mean fit when compared to the multi-class NN, with many also exhibiting better calibration. Meanwhile, on `Reviews`, we find the multi-class NN performs well, though this is also somewhat of a mixed bag: since the cross-entropy approach does not account for ordering, it yields occasional multi-modal pathologies wherein the model places high joint probability on extreme values (i.e., 1 and 5). See Figure 9 for examples.

In general, we favor treating discrete count responses via the natural probabilistic interpretation with an integer-valued random variable. Even in the case where the response is assumed to be finite, we find that a well-fit model learns to decay probabilities for values past the lower and upper bounds (see the case studies in C.2 for a practical example). Additionally, a multi-class model requires the set of labels at train and test time to remain constant. However, in many counting tasks it is plausible that the training data does not cover all possible values.

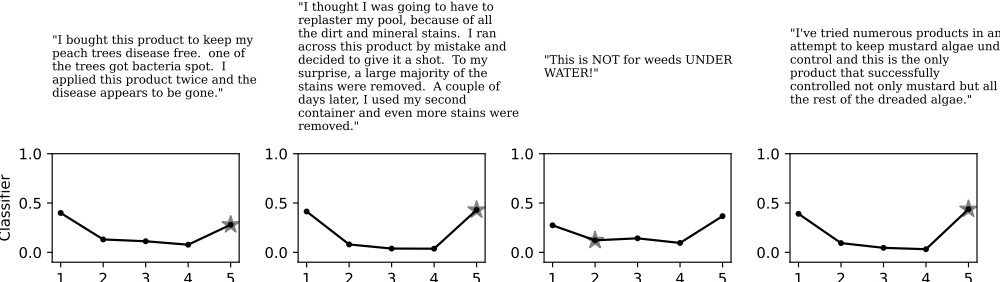

Figure 9: Examples from `Reviews` where a multi-class neural network outputs bimodal distributions. True value of the review is indicated with a star. The lack of a unimodal guarantee is one reason to favor a probabilistic neural regressor over a cross-entropy-based approach.

## C ADDITIONAL CASE STUDIES

### C.1 CASE STUDIES ON COCO-PEOPLE

In this section we perform multiple case studies of the behavior of different heteroscedastic regressors on `COCO-People`. In Figure 10 we display three examples from the `COCO-People` test set and plot the corresponding predictive distributions produced by $\beta_{1.0}$-DDPN. We see varying ranges of predictive uncertainty, while in each case the ground truth count is contained within the predictive HDI.

We next perform a side-by-side comparison of a variety of methods in Figure 11. We display a number of both single forward pass and ensemble methods, plotting their predictive distributions on example images from the test set.

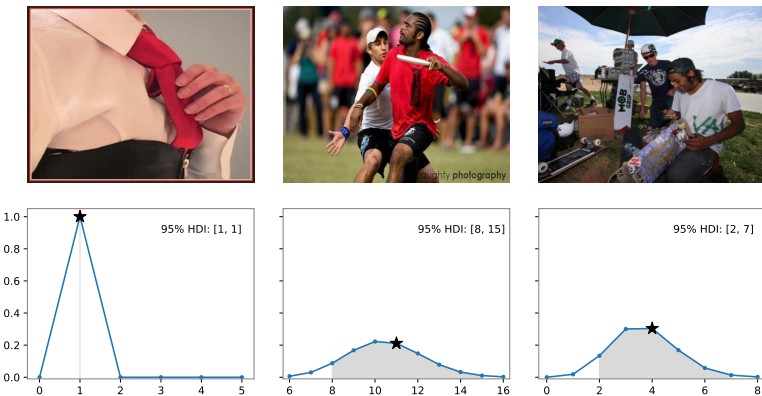

Figure 10: Example $\beta_{1.0}$-DDPN predictive distributions on `COCO-People`. The network is able to flexibly represent counts of different magnitudes with varying degrees of uncertainty, as desired.

### C.2 CASE STUDIES ON AMAZON REVIEWS

In this section we perform a case study of each heteroscedastic method trained on `Amazon Reviews`. We randomly sample four examples from the test split of `Amazon Reviews`. We also sample four random verses from the English KJV Bible. Then, for each method, we plot the predictive distribution of the respective regressor. See Figures 12,13,14,15,16, 17,18, 19, and 20.

A major insight we have from this case study is that, in addition to its strong quantitative performance exhibited in Section 4.4, DDPN appears to provide the best qualitative OOD behavior. In Figure 12

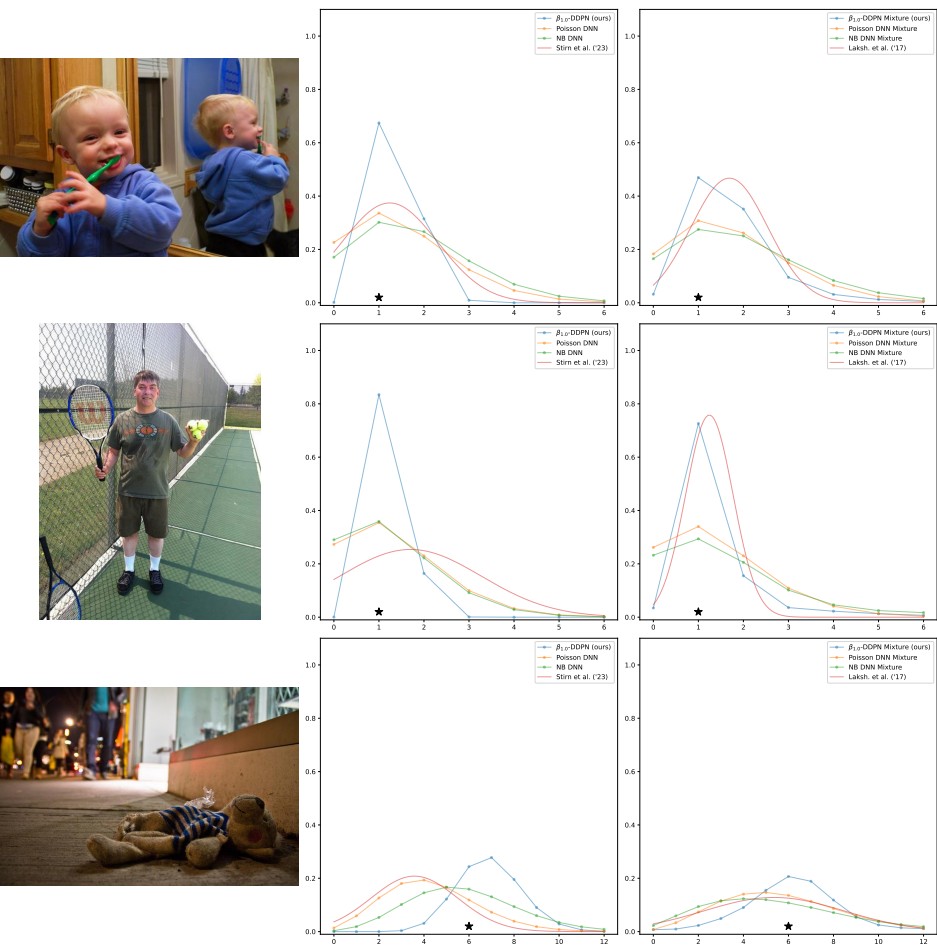

Figure 11: More example predictive distributions on COCO-People. The second column shows distributions output by individual models, while the third column shows outputs from various ensembling techniques. For the sake of visual clarity, for the Double Poisson and Gaussian models, only the best-performing method is shown.

we observe that DDPN exhibits ideal behavior in-distribution with different predictive distributions for reviews with varying valence. However, when fed verses from the KJV Bible, the resulting predictive distributions are essentially the same: diffuse and uninformative across the domain of reviews. In fact, this is evidence that DDPNs revert to the Optimal Constant Solution (OCS) identified by Kang et al. (2023) better than existing methods.

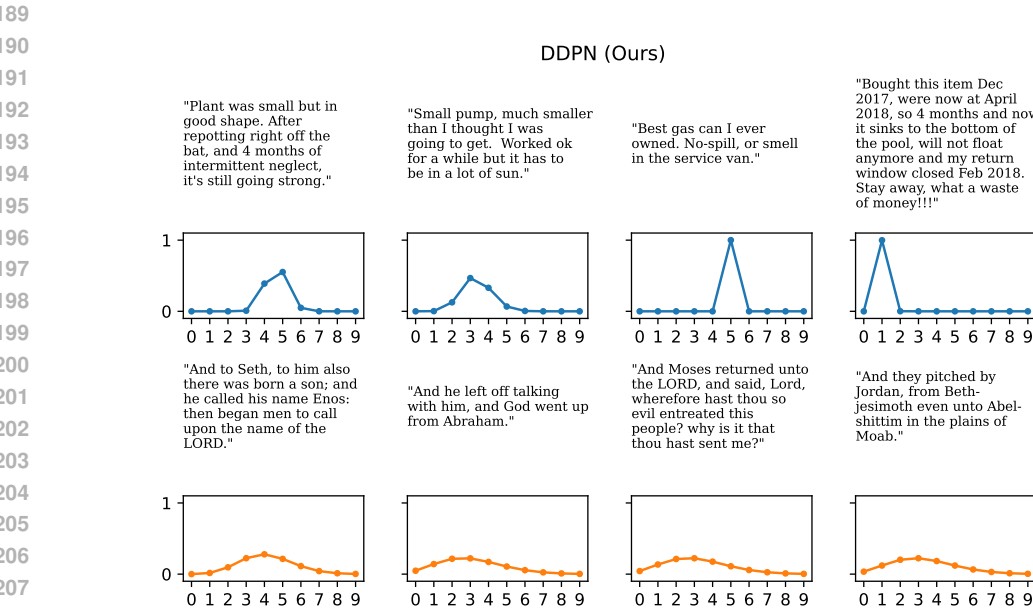

Figure 12: Predictive distributions produced by DDPN on four randomly sampled examples from `Amazon Reviews` and the KJV Bible. DDPN exhibits ideal behavior in-distribution with different predictive distributions for reviews with varying valence. For the KJV Bible, the resulting predictive distributions are essentially the same across examples: diffuse and uninformative. This suggests that DDPNs revert to the Optimal Constant Solution (OCS) identified by Kang et al. (2023) better than existing methods.

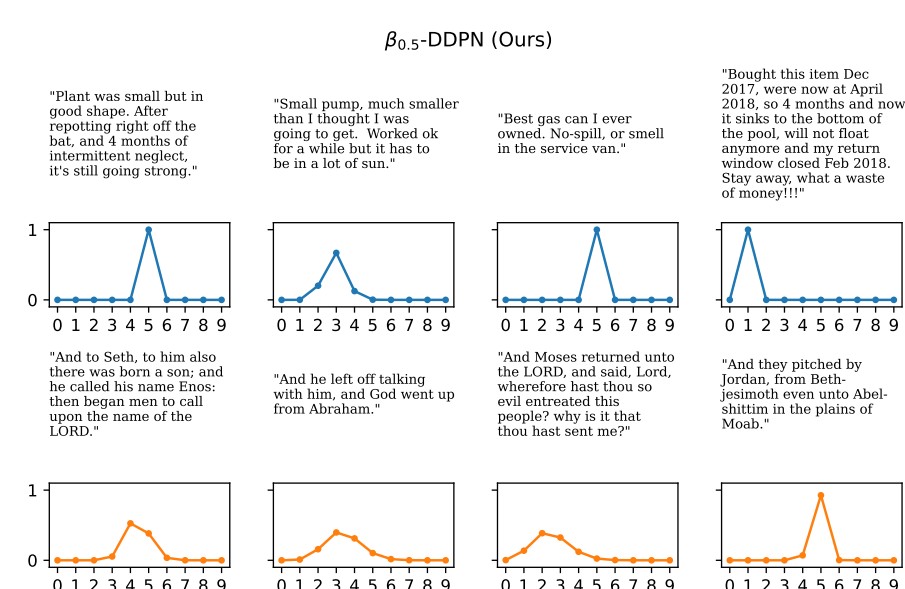

Figure 13: Predictive distributions produced by $\beta_{0.5}$-DDPN on four randomly sampled examples from `Amazon Reviews` and the KJV Bible.

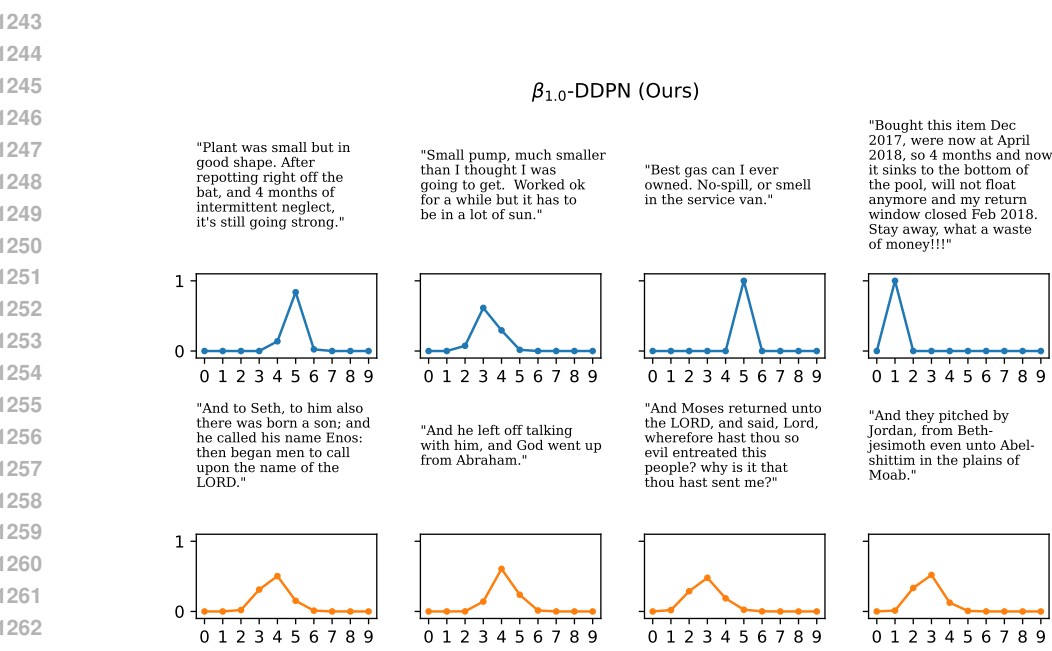

Figure 14: Predictive distributions produced by $\beta_{1.0}$-DDPN on four randomly sampled examples from `Amazon Reviews` and the KJV Bible.

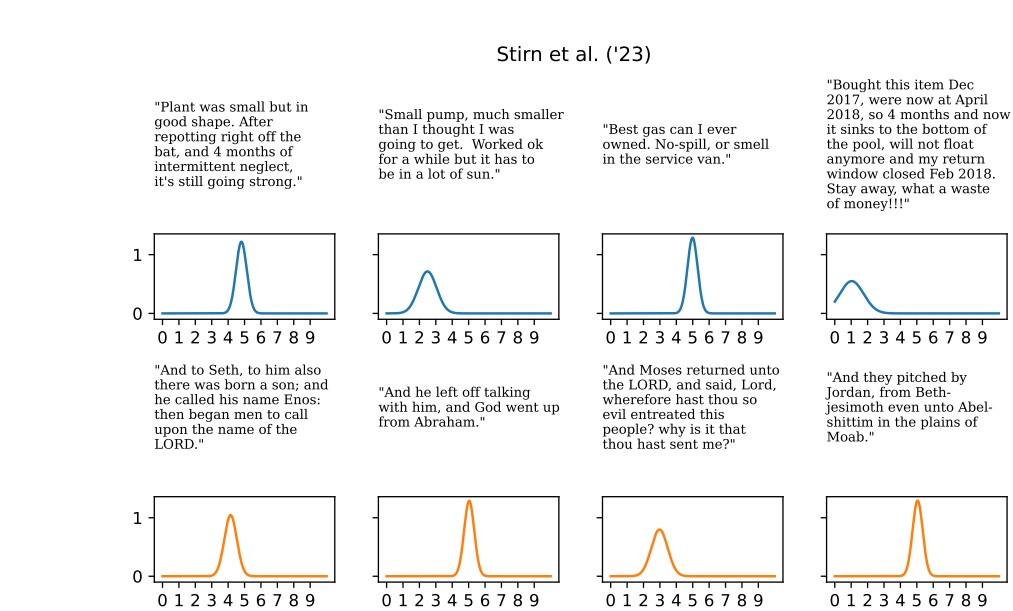

Figure 15: Predictive distributions produced by Stirn et al. (2023) on four randomly sampled examples from `Amazon Reviews` and the KJV Bible.

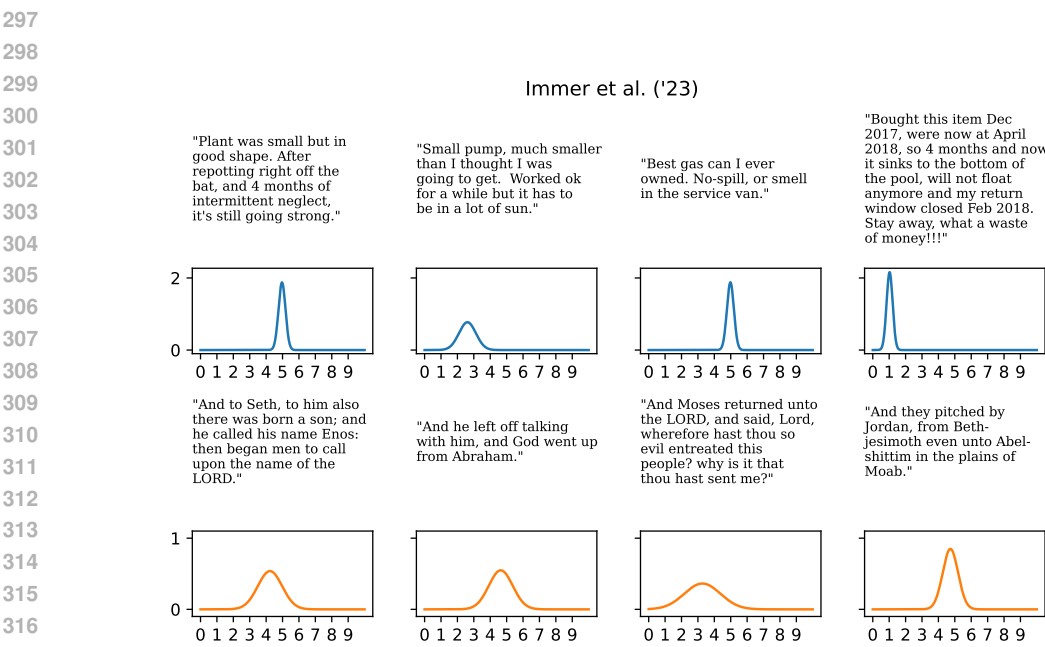

Figure 16: Predictive distributions produced by Immer et al. (2024) on four randomly sampled examples from `Amazon Reviews` and the KJV Bible.

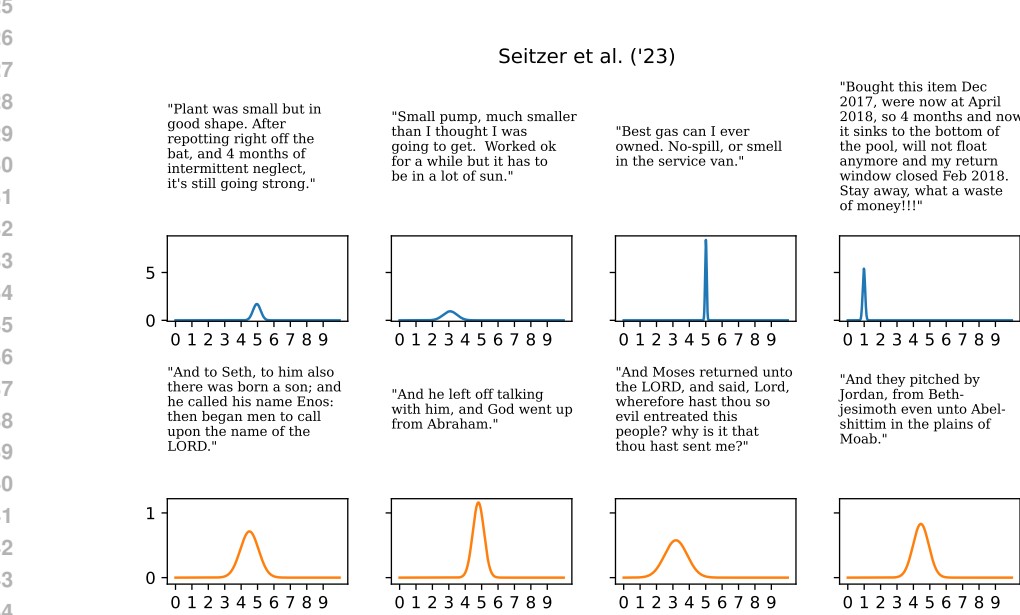

Figure 17: Predictive distributions produced by Seitzer et al. (2022) on four randomly sampled examples from `Amazon Reviews` and the KJV Bible.

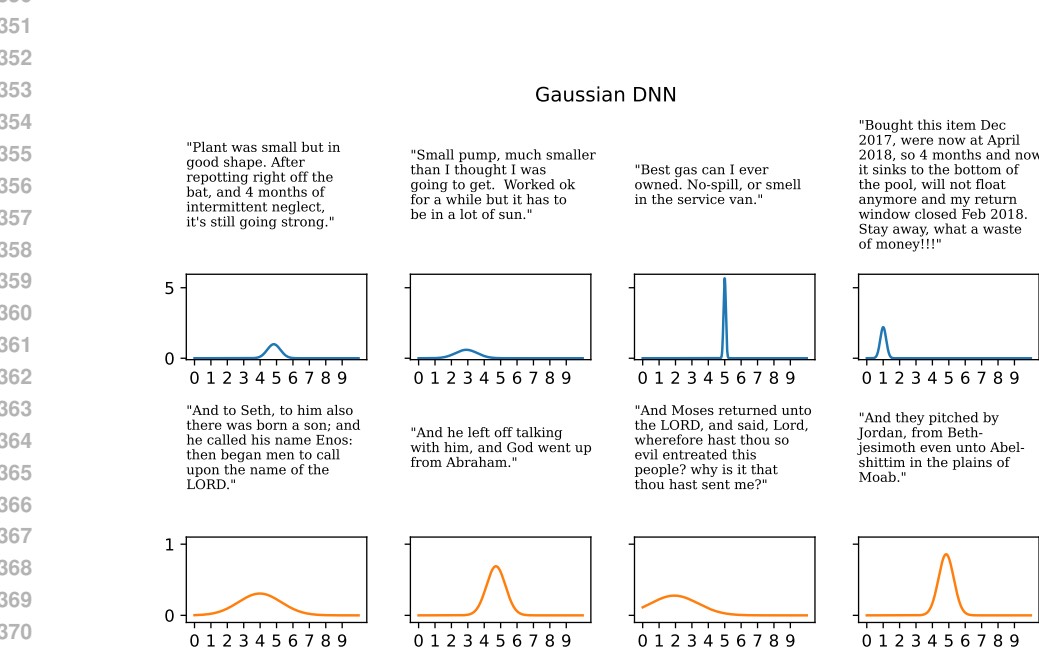

Figure 18: Predictive distributions produced by `Gaussian DNN` on four randomly sampled examples from `Amazon Reviews` and the KJV Bible.

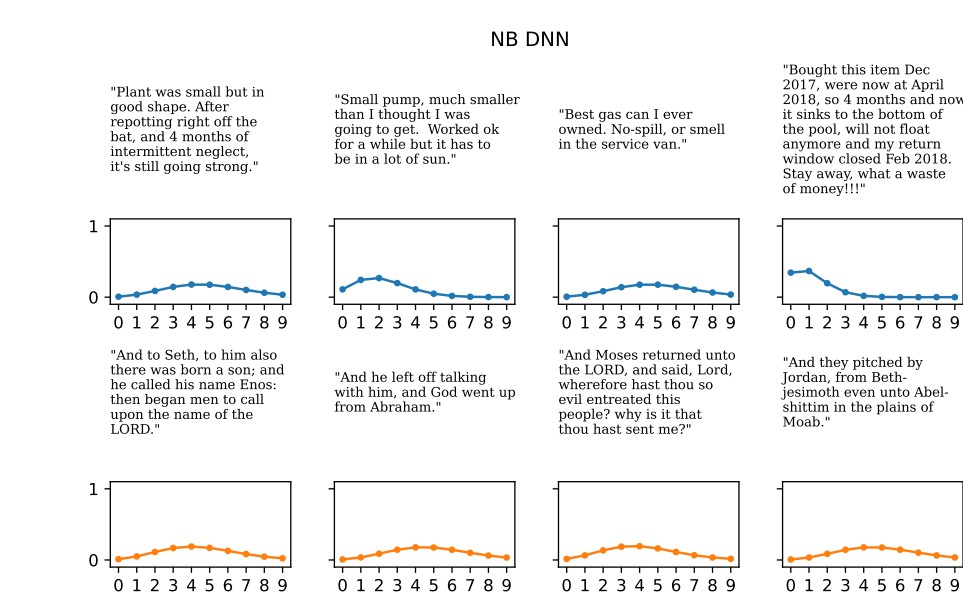

Figure 19: Predictive distributions produced by `NB DNN` on four randomly sampled examples from `Amazon Reviews` and the KJV Bible.

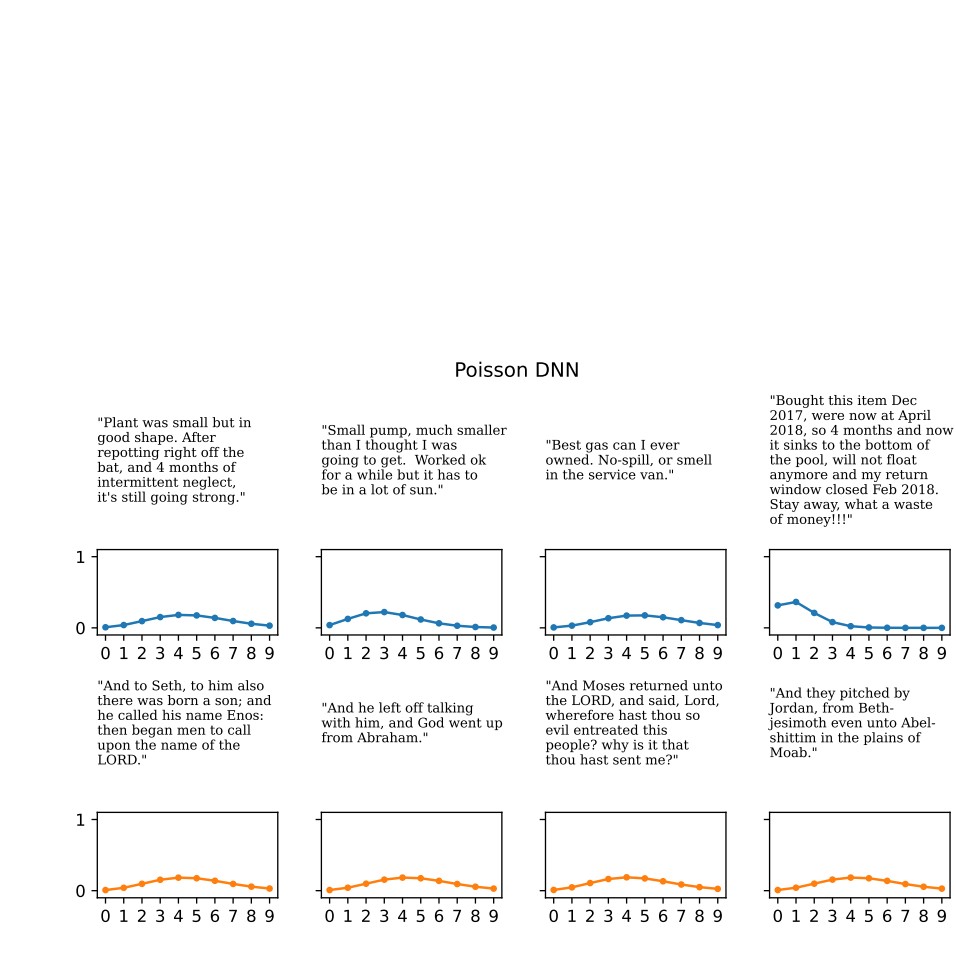

Figure 20: Predictive distributions produced by `Poisson DNN` on four randomly sampled examples from `Amazon Reviews` and the KJV Bible.

## D EXAMPLE POINT CLOUD FROM INVENTORY

In Figure 21, we provide an example point cloud from the `Inventory` dataset used in the experiments of Section 4.3. Further examples can be viewed in Jenkins et al. (2023).

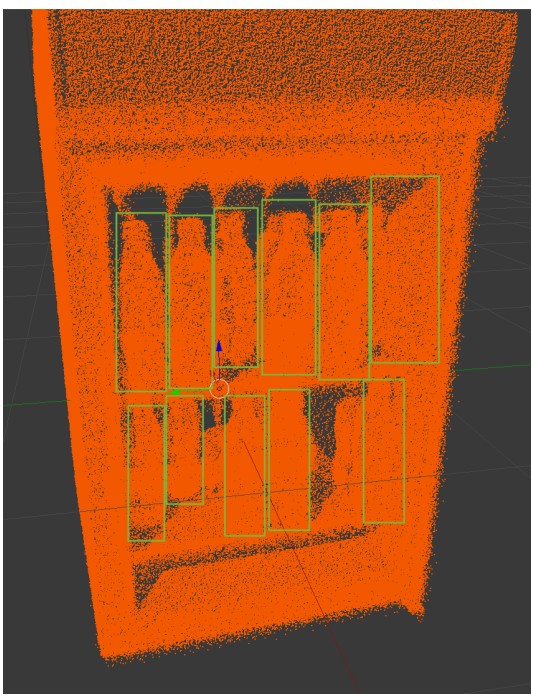

Figure 21: Example point cloud from `Inventory`. Each green box represents an inventory slot which is segmented into a point beam (see Jenkins et al. (2023) for details and further examples). Models predict the product count within each point beam.

| | | COCO-People | Inventory | Amazon | Bikes | Collision |
|---|---|---|---|---|---|---|
| Single Forward Pass | Gaussian DNN | 0.371 (0.04) | 0.704 (0.05) | 7.753 (1.50) | 0.55 (0.09) | 5.424 (1.69) |
| | Poisson DNN | 0.388 (0.04) | 0.252 (0.00) | 0.205 (0.00) | **6.98** (0.08) | 0.871 (0.00) |
| | NB DNN | 0.283 (0.15) | 0.235 (0.03) | 0.205 (0.00) | 1.23 (0.28) | 0.802 (0.04) |
| | Stirn et al. | 0.312 (0.08) | **1.073** (0.13) | **8.789** (0.61) | 2.13 (0.04) | 1.789 (0.06) |
| | Seitzer et al. | 0.432 (0.16) | 0.786 (0.04) | 8.308 (0.97) | 0.96 (0.13) | 6.440 (0.36) |
| | Immer et al. | 0.292 (0.13) | 0.700 (0.02) | 6.671 (1.1) | 0.56 (0.03) | 6.759 (0.45) |
| | DDPN (ours) | 0.366 (0.24) | 0.697 (0.04) | 5.553 (0.30) | 1.39 (0.07) | 7.746 (2.30) |
| | $\beta$-DDPN (ours) | **0.785** (0.37) | 0.745 (0.03) | 8.515 (1.48) | 1.14 (0.14) | **8.343** (0.90) |
| Deep Ensembles | Gaussian DNN | 0.274 | 0.643 | 6.515 | 0.44 | 4.323 |
| | Poisson DNN | 0.278 | 0.244 | 0.205 | **3.97** | 0.863 |
| | NB DNN | 0.124 | 0.225 | 0.205 | 0.93 | 0.799 |
| | DDPN (ours) | 0.194 | 0.641 | 6.632 | 1.15 | 8.567 |
| | $\beta$-DDPN (ours) | **0.296** | **0.664** | **11.30** | 0.92 | **18.228** |

Table 5: Median Precision (MP) across main experiments. We denote the highest value in **bold** and the second-highest with an underline. Note that for the Bikes dataset, all MP values have been multiplied by $10^3$ to lie in a similar scale as other datasets.

# E ADDITIONAL RESULTS FOR REVIEWER DISCUSSION PERIOD

## E.1 ADDITIONAL METRICS FOR MAIN EXPERIMENTS

In this section we provide additional metrics for our main experiments presented in Table 2 and Table 3. We report Median Precision (MP), which is calculated as the median of the precision values, $\lambda_i = \frac{1}{\sigma^2{}_i}$, across the evaluation set. This metric measures the sharpness of the predictive distribution; higher values correspond to more concentrated probability mass. Median precision values are reported in Table 5.

## E.2 ADDITIONAL EXPERIMENTS WITH DEEP ENSEMBLES

We study the performance of ensembling both modern Gaussian regressors (Seitzer, Stirn and Immer) and GLMs. We perform this experiment with all five data sets studied in the main body of the paper. Results for complex data (image, point cloud, and text) are presented in Table 6, and results for tabular data are shown in Table 7.

| | Inventory | | | COCO-People | | | Reviews | | |
|---|---|---|---|---|---|---|---|---|---|
| | MAE | NLL | MP | MAE | NLL | MP | MAE | NLL | MP |
| Seitzer | 0.847 | 1.492 | 0.802 | 2.185 | 2.337 | 0.135 | 0.283 | 0.717 | 8.609 |
| Stirn | 0.878 | 1.552 | 0.907 | 2.384 | 2.519 | 0.117 | 0.282 | 0.740 | 8.482 |
| Immer | 0.881 | 1.529 | 0.622 | 1.917 | 2.183 | 0.263 | 0.277 | 0.678 | 10.203 |

Table 6: Additional deep ensembles trained on complex data. GLMs are omitted because they cannot be easily trained on image, point cloud or text data.

| | Bikes | | | Collisions | | |
|---|---|---|---|---|---|---|
| | MAE | NLL | MP | MAE | NLL | MP |
| Seitzer | 36.755 | 4.953 | 0.000 | 0.274 | 0.722 | 10.774 |
| Stirn | 26.485 | 4.782 | 0.001 | 0.271 | 1.014 | 1.662 |
| Immer | 30.686 | 4.939 | 0.000 | 0.272 | 0.891 | 2.811 |
| Poisson GLM | 109.430 | 9.724 | 0.007 | 0.289 | 1.186 | 0.801 |
| NBinom GLM | 190.026 | 10.772 | 0.613 | 0.290 | 1.188 | 0.804 |
| DP GLM | 189.663 | 7.247 | 0.000 | 0.270 | 0.671 | 7.142 |

Table 7: Additional deep ensembles trained on tabular data

|  | **Bikes** | | | **Collision** | | |
| --- | --- | --- | --- | --- | --- | --- |
|  | $\beta^*$ | MAE ($\downarrow$) | NLL ($\downarrow$) | $\beta^*$ | MAE ($\downarrow$) | NLL ($\downarrow$) |
| Seitzer et al. (2022) | $0.5^3$ | 38.64 (0.80) | 5.01 (0.05) | 0.7 | 0.274 (0.00) | 0.766 (0.09) |
| $\beta$-DDPN | 0.7 | 28.07 (0.53) | 4.67 (0.02) | 0.9 | 0.269 (0.00) | 0.717 (0.02) |

Table 8: $\beta$ tuning results on tabular datasets. We report the optimal value $\beta^*$, the Mean Absolute Error (MAE), and Negative Log Likelihood (NLL) for each method, with standard errors on test metrics derived from 5 separate training/evaluation runs.

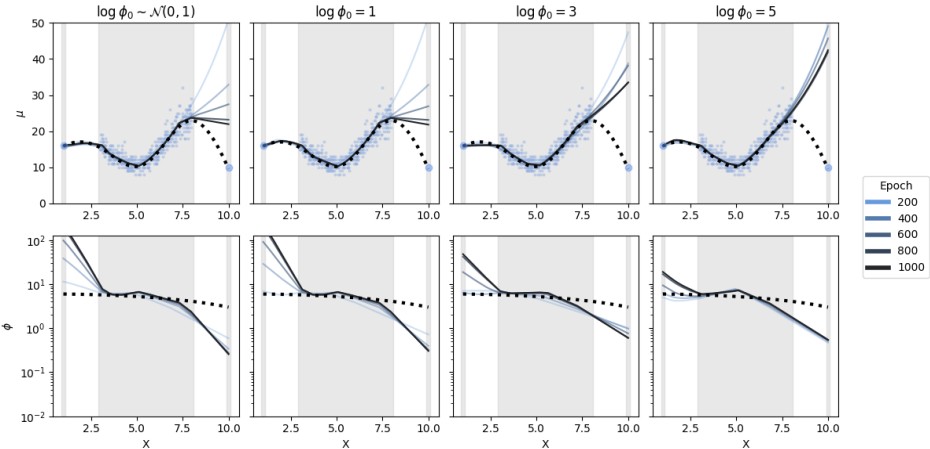

Figure 22: Initialization Experiment 1: What if we initialize $\phi$ for a standard DDPN model to some high value (via the initial bias in the output head)?

### E.3  HOW DOES THE INITIALIZATION OF $\phi$ MEDIATE THE EFFECT OF $\beta$

#### E.3.1  EXPERIMENT 1

What if, for a standard DDPN, we initialize $\phi$ to some high value (via the initial bias in the output head)? Perhaps this helps us avoid the trap of "explaining poor mean fit with high uncertainty" since we're forcing the model to start with low uncertainty values.

Results for this experiment are reported in Figure 22. We see that initializing $\phi$ to a high value at the start of training actually hurts overall convergence to the true function. The best performance comes, in fact, when $\log(\phi)$ is initialized close to zero. Note that despite the high initialization, the point to the far right of the data is still "explained" via low $\phi$ (high uncertainty).

#### E.3.2  EXPERIMENT 2

What is the effect of $\phi$ initialization when training with $\beta$? Does it affect the ability of $\beta$ to steer the model toward the true mean?

Results for this experiment are reported in Figure 23. In this experiment, we set $\beta = 1$. We then initialize the bias on the $\log(\phi)$ prediction layer to 1, 3, and 5, comparing convergence to when$\log(\phi)$ is chosen via the typical standard normal initialization. In contrast to experiment 1, even when we throw the model off with a bad initialization, training with beta helps the model recover and eventually fit the true mean in all but the most extreme cases ($\log(\phi) = 5$).

---

[3]Since $\beta^*$ matches the $\beta$ used in the main experiments, the results have already been obtained and are simply repeated here.

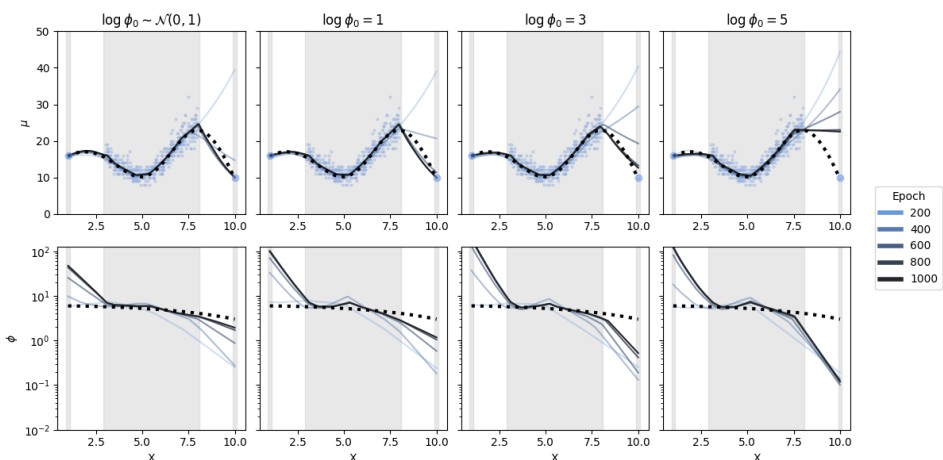

Figure 23: Initialization Experiment 2: What is the effect of $\phi$ initialization when training with $\beta$?

### E.4 SELECTING THE VALUE OF $\beta$

To facilitate a deeper comparison between methods which can be parametrized via a $\beta$ value, we perform the following experiment for both the Bikes and Collision datasets: For $\beta \in \{0.1, 0.2, \ldots, 1.0\}$, we train a Gaussian NN (Seitzer) and a $\beta$-DDPN model. We evaluate each model's performance on the validation split and identify the $\beta$ value that achieves the lowest validation MAE. Using this optimal $\beta^*$, we then train five models, evaluate their performance on the test split, and report the mean and standard deviation of both MAE and NLL. Results are reported in Table 8.

