# OpenReview forum: "Flexible Heteroscedastic Count Regression with Deep Double Poisson Networks"
_ICLR.cc/2025/Conference — Submitted to ICLR 2025_

### Official Review · Reviewer_NZf8 · 2024-10-30

**Soundness:** 3
**Presentation:** 3
**Contribution:** 2
**Rating:** 3
**Confidence:** 4

**Summary:**

This paper presents Deep Double Poisson Network (DDPN), a probabilistic Deep Neural Network (DNN) method by predicting the mean and inverse dispersion of the Double Poisson (DP) distribution for heteroscedastic and discrete count regression problems. DPPN is trained by using the Maximum Likelihood of DP with a “tunable mean fit” parameter $\beta$. The authors show that DDPN and its ensemble version achieve lower Mean Absolute Error (MAE) and Negative Log-Likehood (NLL) than other baselines on regression data and higher predictive entropy in the Out-of-Distribution (OOD) detection setting.

**Strengths:**

- The paper is well-written, and it is easy to understand the important aspects of the algorithm.
- I like the motivation of this paper, the proposed method aims to contribute to discrete regression tasks, such as crowd counting, rating prediction, etc.
- The experimental results show the proposed method is better than some baselines in terms of MAE and NLL on tabular and complex datasets, as well as useful for OOD detection.

**Weaknesses:**

- From my point of view, the proposed method is somewhat not novel enough. Given the literature on the heteroscedastic Gaussian distribution with DNN [1, 2, 3], DDPN simply replaces Gaussian with the DP distribution to tackle the discrete count regression problem.
- The training algorithm depends on the additional tunable parameter $\beta$. There is a trade-off between mean-focused and variance-focused regarding the selection of $\beta$. And, tunning $\beta$ is non-trivial in practice.
- There is no theoretical contribution in this paper. For instance, no theoretical guarantees show that DDPN can achieve better performance (e.g., generalization bound, uncertainty quality bound, etc.) than other methods.
- The experiments are also not convincing. Firstly, DDPN depends on the tunable parameter $\beta$, and its results vary significantly and do not consistently outperform other baselines. Secondly, a lot of measurements to assess model uncertainty quality are missing, e.g., calibration \& sharpness [4], AUPR/AUROC, ROC curve [5], etc.
- Miscellaneous: I feel some sentences are over-claimed. For instance in the abstract, "..4) exhibits superior out-of-distribution detection.". This is not convincing me when seeing the experimental evidence in OOD detection.

**Questions:**

1. In Figure 4, the density across the entropy value of DDPN is uniformly small. Why is this? Can you report the calibration and sharpness value [4] between methods?

2. There is a trade-off between mean-focused and variance-focused regarding the selection of $\beta$. Given the test set is unavailable in the world, how can we choose the best-fit $\beta$ in training to balance this trade-off?

3. Let's consider with only a single forward pass, how DDPN can estimate epistemic uncertainty? And, how DDPN can disentangle aleatoric and epistemic uncertainty?

References:

[1] Lakshminarayanan et al., Simple and scalable predictive uncertainty estimation using deep ensembles, NeurIPS, 2017.

[2] Nix et al., Estimating the mean and variance of the target probability distribution, International Conference on Neural Networks, 1994.

[3] Chua et al., Deep reinforcement learning in a handful of trials using probabilistic dynamics models, NeurIPS, 2018.

[4] Kuleshov et al., Calibrated and sharp uncertainties in deep learning via density estimation, ICML, 2022.

[5] Nado et al., Uncertainty Baselines: Benchmarks for Uncertainty & Robustness in Deep Learning, arXiv preprint arXiv:2106.04015, 2021.

---

> ### Author Response · Authors · 2024-11-22
> **Discussion with Reviewer NZf8**
>
> We thank the reviewer for the thoughtful review and helpful comments! Below is our discussion and answers to key questions.
>
> # Selection of $\beta$ hyperparameter
>
> Tuning can be performed easily with a validation set.
>
> # Lack of theoretical contribution
> We are not claiming strong theoretical contributions. We do provide some theoretical insight into the behavior of the DDPN loss, which is that mean and dispersion can become entangled during training. Specifically, in Equation 3 we show that the partial derivative of the likelihood function strongly depends on the predicted dispersion.
>
> # DDPN does not consistently outperform baselines
> As claimed in the introduction, DDPN significantly outperforms models designed for discrete count problems in nearly every case. DDPN also meets or exceeds the performance of Gaussian-based regressors in nearly every case. Our claim is that DDPN is competitive with Gaussian regressors, but also properly represents probabilities over discrete count data.
>
> # Missing measures (i.e., calibration and sharpness)
>
> We report Median Precision, a measurement of the sharpness of the predictive distribution below. We have added a revised version of the paper and provided a table of MP for all datasets in Appendix E.1.
>
>
> As discussed in the second paragraph in Section 4, we omit ECE because of a known bias when comparing continuous and discrete regressors (Young and Jenkins 2024). Instead, we use NLL since it is a proper scoring rule and accounts for both mean accuracy and calibration. We use the continuity correction to ensure that the comparison between discrete and continuous predictive distributions is fair (See paragraph 2 of section 4).
>
> Spencer Young and Porter Jenkins. On measuring calibration of discrete probabilistic neural networks, 2024. arXiv preprint arXiv:2405.12412
>
> # OOD detection is overclaimed
> In figure 4 a larger $\Delta$ value corresponds to a larger difference between means of the two distributions. If the difference of means is larger, it follows that the central tendency of the distributions is more different. The p-value (from the permutation test) describes how likely this difference is to be statistically meaningful. Therefore, we can see that DDPN has by far the largest delta value, $\Delta=0.460$, demonstrating that it is the best at separating ID and OOD entropy values. This delta value is nearly twice as large as the next best method (Gaussian DNN at $\Delta$=0.238), which is not statistically significant, and more than twice as large as Immer et al, which is statistically significant. For these reasons, we claim that DDPN “exhibits superior out-of-distribution detection.”

---

> > ### Comment · Reviewer_NZf8 · 2024-11-24
> >
> > I thank the authors for the response. While I appreciate your attempt at the rebuttal, my concerns and questions remained. Specifically:
> >
> > **Selection of $\beta$ hyperparameter.**
> > I believe tunning $\beta$ with a validation set is non-trivial, especially when we lack training data. For instance, if we use a hold-out validation set to tune $\beta$, we will sacrifice some training data.
> >
> > **Lack of theoretical contribution.**
> > I don’t mention the authors claimed the paper has strong theoretical contributions. However, to meet the ICLR standards, I believe having a rigorous theoretical guarantee will help ML communities understand how and when the model works.
> >
> > **Miscellaneous about experimental results:**
> > - From the **bold** in Tables 2 and 3, it can be seen that your method does not **consistently** outperform other baselines.
> >
> > - As mentioned, I believe NLL is not sufficient to evaluate uncertainty quality. It does not reflect the difference in expectation between model accuracy and certainty. It also does not reflect the width of the model’s confidence interval.
> >
> > My other questions are also not fully addressed. I keep my original rating for this paper.

---

> > > ### Author Response · Authors · 2024-11-25
> > > **Additional Discussion with NZf8**
> > >
> > > # Hyperparameter Tuning
> > >
> > > While we share the reviewer's concerns about sacrificing some training data to tune $\beta$, this is a common and widely-accepted tradeoff in machine learning. Another alternative for hyperparameter selection could be to use cross-validation, although we note that this approach is computationally intensive (assuming k folds and m choices of $\beta$ across a grid, we now must train and evaluate $mk$ total models, which can grow computationally prohibitive with larger data/models).
> > >
> > > Due to the difficulties and additional compute required to perform proper hyperparameter tuning, we also provide some intuition regarding choosing the value of $\beta$ in the final paragraph of Section 3.2:
> > >
> > > > The Double Poisson β-NLL is parameterized by β ∈ [0, 1], where β = 0 recovers the original Double Poisson NLL and β = 1 corresponds to fitting the mean, μ, with no respect to ϕ (while still performing normal weight updates to fit the value of ϕ). Thus, we can consider the value of β as providing a smooth interpolation between the natural DDPN likelihood and a more mean-focused loss (Figure 3). For an empirical demonstration of the impact of β on DDPN, see Figure 5.
> > >
> > > Thus, we present a principled framework for choosing $\beta$. We hope that these clarifications are satisfactory to the reviewer.
> > >
> > > # Experimental Results: Performance Consistency
> > >
> > > We acknowledge that DDPN does not achieve absolute dominance across all datasets. However, our claims focus on DDPN's consistent competitiveness across discrete count tasks, where it often exceeds Gaussian-based regressors in NLL and MAE (while consistently outperforming Poisson or Negative Binomial-based regressors). For example, in Tables 2 and 3, DDPN (or $\beta$-DDPN) achieves the lowest NLL in 70% of cases and matches or exceeds the best-performing baseline in MAE across 80% of cases. These results underline DDPN's practical utility in discrete regression.
> > >
> > > # On Estimating Epistemic Uncertainty
> > >
> > > We apologize to the reviewer for neglecting to respond to question 3 in your original review. DDPN as proposed cannot estimate epistemic uncertainty, nor do we claim it can. DDPN is a technique for estimating aleatoric uncertainty as discussed in Section 2.2. DDPN can be combined with common epistemic methods such as deep ensembles (Section 3.3).

---

### Official Review · Reviewer_xkAG · 2024-11-03

**Soundness:** 2
**Presentation:** 3
**Contribution:** 2
**Rating:** 5
**Confidence:** 3

**Summary:**

This work attempts to improve discrete heteroscedastic regression by predicting the parameter for a double poison distribution.

The contributions include: 1) the method outperforms the baselines on some datasets; 2) the authors modify the NLL loss to leverage between mean and distribution prediction; 3) the method exhibits superior out-of-distribution detection functionality.

**Strengths:**

The paper is easy to follow, with a clear presentation.

The proposed method:
1) shows superior results regarding MAE or NLL on some of the datasets.
2) provides the $\beta$ modification to the loss to enable prioritizing either the mean or the distribution prediction.
3) exhibits superior OOD detection behavior.
4) shows how the $\beta$ trick effects the convergence of DDPN.

**Weaknesses:**

The novelty of the paper: the method changes the output distribution compared to previous work. For example, while previous work predicts the parameter for a Gaussian distribution, this work predicts the parameter for a Double Poison distribution. However, the connection between the new distribution and the superior performance is not clearly analyzed. It remains possible that one can try different distributions for different datasets. Thus, the method is not innovative from the methodology perspective. Other than the distribution change, the method also relies on the $\beta$ trick proposed by Seitzer et al, which further reduces the originality of the work.

Experiment results: while the method shows superior performance for one of the metrics in most cases, it does not necessarily indicate the significance of the model. Instead, tuning and applying the $\beta$ trick to some of the baseline methods may lead to better performance. However, it is not explored. Thus, it remains unclear whether it is the introduction of a new distribution or the $\beta$ trick that leads to the improvement. I recommend the author to further explore the trick with baseline methods and make a fair comparison to show what is behind the improvement.

For the OOD detection, the DDPN and variants indeed show better behavior compared to some of the results. However, it does not seem to be better than Immer et al.

**Questions:**

1. Could you explain why Double Poison is more appropriate compared to the other distributions? Or in which case, this distribution should show superior performance?

2. Could you show how $\beta$ affects the performance of the baseline methods and explore them also in the same space ($\beta \in [0,1]$)?

3. Are DDPN and the variants better in OOD detection compared to Immer et al? It is not obvious from Fig.4. Could you further illustrate the reason why you claim "DDPN shows the greatest ability of all benchmarked regression models to differentiate better ID and OOD inputs"?

4. Could you explain why some of the baselines are missing from the boosting experiments?

---

> ### Author Response · Authors · 2024-11-23
> **Discussion with Reviewer xkAG**
>
> We thank the reviewer for the thoughtful review and helpful comments! Below is our discussion and answers to key questions.
>
> # Lack of significance of experimental results
>
> As claimed in the introduction, the base DDPN significantly outperforms models designed for discrete count problems in nearly every case. Additionally, the base DDPN meets or exceeds the performance of Gaussian-based regressors in nearly every case. Our claim is that DDPN is competitive with Gaussian regressors, but also properly represents probabilities over discrete count data. This claim is supported by our experiments.
>
> The introduction of $\beta-DDPN$ is to give tunable preference to mean fit or calibration. $\beta-DDPN$ meets or exceeds the performance of its Gaussian counterpart proposed by Seitzer, but also properly represents probabilities over discrete count data.
>
> # OOD experiment: comparison to Immer et al.
>
> In figure 4 a larger $\Delta$ value corresponds to a larger difference between means of the two distributions. If the difference of means is larger, it follows that the central tendency of the distributions is more different. The p-value (from the permutation test) describes how likely this difference is to be statistically meaningful. Therefore, we can see that DDPN has by far the largest delta value, $\Delta=0.460$, demonstrating that it is the best at separating ID and OOD entropy values. This delta value is nearly twice as large as the next best method (Gaussian DNN at $\Delta$=0.238), which is not statistically significant, and more than twice as large as Immer et al, which is statistically significant. For these reasons, we claim that DDPN “exhibits superior out-of-distribution detection.”
>
> # Why is Double Poisson more appropriate  compared to other distributions?
> The core problem is that we wish to model probabilities over discrete count data.
>
> The Gaussian distribution is flexible, but is a continuous distribution, and therefore is inappropriate for count data. We identify three pathologies with simply applying the Gaussian to count data in Section 1:
>
> > First, the continuous predictive distribution will assign non-zero probability mass to infeasible real values that fall in between valid integers. Second, the predictive intervals are unbounded and can assign non-zero probability to negative values when the predicted mean is small. Third, the boundaries of the predictive intervals (i.e., high density interval or 95% credible interval) are likely to fall between two valid integers, diminishing their interpretability and utility.
>
>
> DDPN resolves all three of these pathologies
>
> Existing discrete likelihood models produce proper distributions over count data, however they are rigid and cannot model under-, equi-, and over-dispersion. DDPN is more flexible (resulting in better performance) and can model all of three of these cases.
>
> # Missing baselines from 'boosting' experiments
>
> While we do not perform any experiments with boosting, we believe the reviewer is referring to our experiments with Deep Ensembles. To the best of our knowledge, Deep Ensembles with very recent Gaussian regressors (Immer et al., Stirn et al., and Setizer et al.) have never appeared in the literature, and are therefore completely new baselines and were not included in the original manuscript. To address the reviewer's concern have added a revision and now include these results in Appendix E.2. In all cases, the DDPN ensemble outperforms these additional baselines in terms of MAE.

---

> > ### Author Response · Authors · 2024-11-24
> >
> > # On tuning the value of $\beta$
> >
> > We appreciate the reviewer’s insightful suggestion to investigate the impact of $\beta$ tuning across baseline methods. However, the $\beta$ modification, as introduced in Section 3.2, is designed specifically to address a gradient phenomenon observed in models predicting both a mean and a dispersion term (e.g., $\sigma$ for Gaussian NN, $\phi$ for DDPN). This phenomenon, which is identified via partial derivative analysis, hinders the model’s ability to fit the mean during gradient descent due to getting "stuck" explaining poor mean fit via high uncertainty. Our discovery of this gradient issue in the context of Double Poisson NLL is a novel contribution, as prior work has only analyzed and applied this insight to Gaussian NLL.
> >
> > ## Applicability of $\beta$ to Baselines
> >
> > Not all baseline methods are suitable candidates for the $\beta$ modification:
> >
> > 1. Gaussian methods (Immer and Stirn): These approaches already address the mean fit issue through alternative mechanisms (see Section 2.2.2, paragraph 1), rendering a $\beta$ modification redundant and potentially counterproductive.
> > 2. Methods using Poisson or Negative Binomial heads: These baselines inherently avoid the gradient dynamics that necessitate $\beta$, as their loss functions do not depend on a dispersion parameter in the same way.
> >
> > Thus, the $\beta$ trick is applicable only to the Gaussian NN (Seitzer’s method) and DDPN. We have already included results in the paper for DDPN with $\beta = 0$ (standard DDPN), $\beta = 0.5$, and $\beta = 1.0$, as well as for the Gaussian NN with $\beta = 0$ (standard Gaussian NN) and $\beta = 0.5$ (Seitzer’s recommended setting).
> >
> > ## New Experiment: Tuning $\beta$ for Gaussian NN and DDPN
> >
> > To address the reviewer’s concern about the potential advantage conferred by $\beta$ tuning, we conduct additional experiments on the bikes and collisions datasets. For $\beta \in {0.1, 0.2, \dots, 1.0}$, we train Gaussian NN (Seitzer) and $\beta$-DDPN models, evaluate their performance on the validation split, and identify the $\beta$ value that achieved the lowest validation MAE. Using this optimal $\beta$, we train five models, evaluate their performance on the test split, and report the mean and standard deviation of MAE and NLL.
> >
> > The results, presented in Appendix E.4 of our revised manuscript, demonstrate that $\beta$ tuning can lead to further improvements (or roughly equivalent performance) over the metrics achieved by DDPN in our original experiments. Note that, for both datasets, the tuned $\beta$-DDPN outperforms the tuned Seitzer model in terms of both MAE and NLL. This performance edge highlights the advantage of the Double Poisson distribution: it better incorporates domain knowledge when regression targets are nonnegative integers.
> >
> > ## Ongoing Work
> >
> > Due to time constraints during the discussion period, we focused our tuning experiments on two datasets (bikes and collisions). We are actively extending this analysis to the remaining benchmark datasets and will include the complete results in the camera-ready version of the paper if accepted.

---

### Official Review · Reviewer_G7zs · 2024-11-04

**Soundness:** 1
**Presentation:** 2
**Contribution:** 2
**Rating:** 3
**Confidence:** 4

**Summary:**

The authors proposed DDPN for heteroscedastic count data along with a $\beta$-modification to enhance prediction performance. While the introduction of heteroscedasticity into neural networks and the focus on discrete data are appreciated, there is a lack of investigation into existing methods, and one of the two presented measures seems not suitable for comparing the methods.

**Strengths:**

The introduction of heteroscedasticity into neural networks is a notable strength, as it enables more flexible and accurate modeling of data with varying dispersion. Additionally, the focus on discrete data addresses a gap in existing research, highlighting the method's practical relevance for count-based outcomes.

**Weaknesses:**

**1. Insufficient discussion of existing methods**

The authors identify the limitations of several approaches to representing heteroscedasticity in Poisson distributions, providing motivation for their proposed method. However, the authors did not mention Joint GLM, which has been widely discussed in the literature. For example, Chapter 10 of *Generalized Linear Models* (McCullagh and Nelder, 1989)—which the authors cited as McCullagh (2019)—discusses "Joint modeling of mean and dispersion" within the GLM family. *Generalized Linear Models with Random Effects* (Lee, Nelder, and Pawitan, 2017) also examines Joint GLM for mean and dispersion, which is a natural extension of heteroscedastic model for Gaussian cases to the GLM family. As far as I know, linear components of Joint GLM can be replaced with neural networks, but the authors did not clarify how the use of the Double Poisson distribution offers distinct advantages over this approach.

**2. Concerns with comparing likelihoods across different distributions**

There is a fundamental issue with the presentation. The authors directly compared likelihood values under different distributional assumptions. However, such comparisons are strongly discouraged as they can lead to misleading conclusions. The authors should acknowledge this and include the necessary methodological context or justification.

**3. Need for theoretical support and further investigation of $\beta$-DDPN**

The introduction of the $\beta$-DDPN and its demonstration in Figure 5 is interesting, but the method currently lacks sufficient theoretical support. A more comprehensive and foundational investigation into convergence could provide deeper insights. For instance, an ablation study on the bias and variance of weight estimates in linear models could help clarify the influence of the $\beta$ value on convergence behavior.

**Questions:**

**1.  On Existing Methods:**
- The authors omitted a discussion on Joint GLM, which is widely acknowledged in the literature and even appears in McCullagh (2019), cited by the authors. Clarification on this omission would be required.
- Could the authors elaborate on how the Double Poisson distribution provides distinct advantages over Joint GLM?

**2. On Comparing Likelihoods:**
- Could the authors justify comparing likelihood values across different distributional assumptions?
- If such justification is not possible, considering alternative measures for comparison would be required.
- For certain datasets, differences in MAE does not seem significant. Providing statistical test results would support the analysis.

**3. On Theoretical Support for $\beta$-DDPN:**
- An ablation study on the bias and variance of weight estimates in simpler (linear) models could provide insights into the impact of the $\beta$ value on convergence.
- It seems that the influence of the $\beta$ value might depend on whether the initial value of $\phi$ is set high or low. Could the authors explain or provide insight into this potential dependency?

---

> ### Author Response · Authors · 2024-11-23
> **Discussion with Reviewer G7zs**
>
> We thank the reviewer for the thoughtful review and helpful comments! Especially for the opportunity to discuss and clarify connections of DPPN to previous work. Below is our discussion and answers to key questions.
>
>
> # Connections to Joint GLM
> The original Double Poisson GLM presented by Efron 1986 is in fact a joint GLM as it jointly models mean and dispersion of a response variable. McCullagh and Nelder, 1989 define the joint GLM in the following form:
>
> $\eta_i = g(\mu_i) = \boldsymbol{x}_i^T\boldsymbol{\beta}$
>
> $\zeta_i = h(\phi_i) = \boldsymbol{u}_i^T \boldsymbol{\delta}$
>
> Where $E(d(Y_i, \mu_i)) = \phi_i$, and $d(Y_i, \mu_i)$ is a dispersion statistic, which depends on the mean. Here, the covariates, $\boldsymbol{x}_i$ and $\boldsymbol{u}_i$, can either be shared or be different.
>
> This model has two apparent defining properties: 1) that the first two moments depend on covariates; and 2) the dispersion depends on some way on the mean, $\mu_i$
>
> In Section 2.2.1 of our manuscript, we define the DP GLM using the same specification as Efron 1986:
>
> Given parameter vectors $\boldsymbol{\alpha}$ and $\boldsymbol{\beta} = [\beta_0, \beta_1, \beta_2]^T$,
> let $\text{log}(\hat{\mu_i}) = \hat{\eta_i} = \boldsymbol{\alpha}^T\boldsymbol{x}_i$ and $\hat{\sigma}_i = \frac{M}{1 + e^{-(\beta_0 + \beta_1 \hat{\mu}_i + \beta_2 \hat{\mu}_i^2)}}$
>
> Thus, it is apparent that the DP GLM baseline in our experiments is in fact a joint GLM because 1) DP GLM parameterizes the first two moments as a function of covariates, and 2) the dispersion depends on the mean.  In the same section, we discuss limitations of this approach
>
> > This approach has two key limitations: 1) the predicted dispersion, $\hat{\sigma}_i$, does not directly depend on the input $\boldsymbol{x}_i$, and is instead a function of the predicted mean, $\hat{\mu}_i$); 2) the hyperparameter $M$ introduces an upper bound on the dispersion, which in turn curtails the feasible range of confidence values. In practice, the authors set $M=1.25$, which hardly allows for under-dispersion ($\sigma > 1$).  Both of these measures significantly limit the family of distribution functions the model can learn. Follow-up studies all assume a constant dispersion term, applying even stronger limits on flexibility. In contrast to these, our proposed approach drastically expands the family of functions that can be modeled. DDPN learns a non-linear mapping that can be trained on complex data and can fully disentangle the mean and dispersion, allowing for pure heteroscedastic regression. We also introduce a tunable hyperparameter that allows for custom prioritization between mean fit and overall likelihood calibration.
>
> We also argue that DDPN offers the following advantages:
>
> > in contrast to these, our proposed approach drastically
> expands the family of functions that can be modeled. DDPN learns a non-linear mapping that can be trained on complex data and can fully disentangle the mean and dispersion, allowing for pure heteroscedastic regression. We also introduce a tunable hyperparameter that allows for custom prioritization between mean fit and overall likelihood calibration.
>
> Additionally, we show empirically that DDPN improves over the Joint GLM in Table 2 for two tabular datasets.
>
>
> # Comparing likelihoods and computing calibration
>
> As discussed in the second paragraph in Section 4, we omit ECE because of a known bias when comparing continuous and discrete regressors [1]. Instead, we use NLL since it is a proper scoring rule and accounts for both mean accuracy and calibration. We use the continuity correction [2] to ensure that the comparison between discrete and continuous predictive distributions is fair (See paragraph 2 of section 4).
>
> [1] Spencer Young and Porter Jenkins. On measuring calibration of discrete probabilistic neural networks, 2024. arXiv preprint arXiv:2405.12412
>
> [2] "Continuity Correction." Wikipedia, Wikimedia Foundation, 23 Nov. 2024, https://en.wikipedia.org/wiki/Continuity_correction.
>
> # Significance of MAE
> As claimed in the introduction, DDPN significantly outperforms models designed for discrete count problems in nearly every case in terms of MAE. Additionally, the DDPN meets or exceeds the MAE performance of Gaussian-based regressors in nearly every case. In some cases, the MAE differences may be within the standard error of the MAE reported, resulting in a statistical tie. This is still consistent with our claim that DDPN is competitive with Gaussian regressors, but also properly represents probabilities over discrete count data.
>
>
> # How does the initialization of $\phi$ mediate the effect of $\beta$?
>
> We have added a revision and include two experiments studying this question in Appendix E.3.
>
> Experiment 1: What if we initialize phi to some high value (via the initial bias in the output head)?
>
> Experiment 2: What is the effect of phi initialization when training with beta? Does it affect the ability of beta to steer the model toward the true mean?

---

> ### Comment · Reviewer_G7zs · 2024-11-25
>
> I thank the authors for their efforts in providing detailed responses to our comments. However, several critical points remain unresolved. Thus, I keep my original rating for this paper.
>
> ### Regarding the relationship with Joint GLMs
> - While both Joint GLMs and DP GLMs model the frist two moments, this alone does not imply that the two models are identical. For instance, the authors noted that $\text{Var}(Y_i)$ is bounded by the hyper-parameter $M$ in DP GLMs, whereas such a bound does not generally apply in Joint GLMs.
> - Joint GLMs model the orthogonal dispersion parameter $\phi_i$ rather than directly modeling $\text{Var}(Y_i)$. In the standard exponential dispersion model framework, the variance typically takes the form $\text{Var}(Y) = \phi \cdot V(\mu)$. In the case of Gaussian distribution, $\text{Var}(Y_i)$ can be directly modeled since $V(\mu_i)=1$. However, for Poisson distributions, $\mu_i$ and $\text{Var}(Y_i)$ would not be orthogonal, since $V(\mu_i)=\mu_i$. As far as I understand, this is why joint GLMs typically model $\phi_i$ rather than $\text{Var}(Y_i)$. Therefore, without further justification, the authors' claim regarding the first limitation could be perceived as unconvincing and potentially overstated from certain perspective.
>
> ### Comparing NLLs from different distributions
> - Applying a continuity correction may not justify directly comparing the NLL values from different distributional assumptions, as it goes against fundamental principles of likelihood theory.

---

> > ### Author Response · Authors · 2024-11-25
> > **Comparing NLLs from different distributions**
> >
> > We sincerely thank the reviewer for taking the time to help us think through the relationship with Joint GLMs. This discussion will no doubt improve future drafts of the paper.
> >
> > We are facing a difficult challenge in evaluating the quality of predictive uncertainty for both discrete and continuous models. Clearly, the reviewer has problems with the NLL + continuity correction. We have reason to believe from recent work [1] that ECE is biased towards continuous regressors, when compared to discrete regressors.
> >
> > **Question:** How would the reviewer suggest we quantitively the compare the predictive distributions of discrete and continuous regressors in our experiments?
> >
> >
> > [1] Spencer Young and Porter Jenkins. On measuring calibration of discrete probabilistic neural networks, 2024. arXiv preprint arXiv:2405.12412. See Appendix D.1

---

### Official Review · Reviewer_dvqi · 2024-11-04

**Soundness:** 3
**Presentation:** 3
**Contribution:** 3
**Rating:** 8
**Confidence:** 4

**Summary:**

This paper introduces Deep Double Poisson Networks (DDPN) for heteroscedastic regression on discrete count data.
The key contributions is using the double Poisson distribution as the output distribution, which provides more flexibility than Poisson or negative binomial distributions while maintaining proper support over integers (unlike Gaussian which has pathologies behavior because of this).
The authors also propose a $\beta$-modification to the loss function to better balance mean fit and uncertainty calibration.
Comprehensive experiments across tabular, image, point cloud and text data demonstrate DDPN's superior performance in terms of both prediction accuracy and uncertainty quantification.

**Strengths:**

- Clear motivation and principled solution to discrete regression with uncertainty estimation
- Strong theoretical analysis of the relationship between different discrete distributions and why double Poisson is more flexible
- Comprehensive empirical evaluation across diverse datasets and modalities
- Excellent ablation studies demonstrating the impact of the $\beta$ parameter
- Strong out-of-distribution detection capabilities compared to baselines
- Good reproducibility with code provided

**Weaknesses:**

- Some empirical results (e.g., on Amazon review dataset) show relatively modest improvements over strong baselines
- Could benefit from more analysis of when simpler distributions might be sufficient

**Questions:**

1. How sensitive is the method to the choice of $\beta$? Are there heuristics for selecting it for different applications?
2. Could you elaborate on why $\beta$=0.5 seems to work particularly well for ensembles?

---

> ### Author Response · Authors · 2024-11-14
> **Discussion with reviewer dvqi**
>
> We thank the reviewer for the thoughtful review and helpful comments! Below is our discussion and answers to key questions.
>
> # Modest Improvements over baselines on Amazon dataset
>
> While some of the more modern Gaussian regressors (i.e., Stirn, Seitzer, Immer), are indeed competitive with DDPN, they are still subject to the pathologies and mis-specification issues discussed in Section 1. Our claimed contributions are not that DDPN outperforms all baselines all of the time. Rather, we claim that DDPN can meet or exceed the performance of modern Gaussian regressors, and also produce a properly specified predictive distribution over discrete count data, which the Gaussian does not. If we truly believe that the data are discrete, we should incorporate that belief into our modeling approach.
>
> # Comparison to simpler distributions
>
> In all of our experiments we compare to simpler distributions (Table 2, Table 3). Specifically, we compare DDPN to the Poisson DNN and the Negative Binomial (DNN), which are both simpler distributions. Poisson has a single parameter for mean and variance, while the NB DNN has two parameters, but is restricted to variance >= mean, and is therefore less flexible.
>
> We see on tabular data with naturally high dispersion conditioned on the observations (i.e., Bikes in Table 2), Poisson is comparable to DDPN. However, on lower dispersion Tabular data (i.e., Collision in Table 2) and in the more high dimensional data in Table 3, DDPN outperforms existing discrete approaches. In general, DDPN can fit data of all dispersions, even when technically “misspecified” as we demonstrate in Appendix A.2.
>
> # Sensitivity to choice of $\beta$
>
> In appendix A.4 we perform a hyperparameter study of the effect of Beta on the COCO-people dataset.
>
> # Why does $\beta=0.5$ seem to work particularly well for ensembles?
>
> It is possible that $\beta=0.5$ yields more diverse independently trained models. Thus we capture a greater variety of plausible minima in weight space. In this scenario each mode's mistakes cancel the others out, resulting in a stronger ensemble. If $\beta=1$, it’s likely that the models all tend towards similar solutions where they fit the mean with high (sometimes too-high) confidence. Conversely, if $\beta=0$ the models all tend towards similar solutions with high uncertainty explaining away poor mean fit. A follow-up work should study this more in-depth.

---

> > ### Author Response · Authors · 2024-11-16
> > **Empirical results describing the effect of $\beta$ on ensemble diversity**
> >
> > To investigate the hypothesis that $\beta=0.5$ creates more diverse members of an ensemble, we performed the following experiment with the inventory dataset: for DDPN models trained with $\beta=0.0$, $\beta=0.5$, and $\beta=1.0$, we recorded the probabilistic predictions of each individual ensemble member along with the overall ensemble (mixture) prediction. We then computed the average mutual information between the ensemble and the individual members as specified in Equation 3 of [1]. As noted in [1], this provides a notion of ensemble diversity, since it measures "the mean KL-divergence between the predictive distributions of the ensemble members and the ensemble overall" (for each data point). Our results are as follows (averaged over all data points):
> >
> >
> > | $\beta$ | Mutual Information |
> > |----------|----------|
> > | 0.0   | 0.0457 |
> > | 0.5   | 0.0915 |
> > | 1.0.  | 0.0614 |
> >
> >
> > We observe that the highest mean KL-divergence occurs when ensemble members are trained with $\beta=0.5$. This provides evidence that this setting yields more diverse models when independently trained.
> >
> > [1] Xia, G., & Bouganis, C. S. (2022). On the usefulness of deep ensemble diversity for out-of-distribution detection. arXiv preprint arXiv:2207.07517.

---

### Meta-Review · Area_Chair_w9Rx · 2024-12-20

**Metareview:**

The paper introduces Deep Double Poisson Networks (DDPNs) for count regression, which are capable of modeling under-dispersed, equal-dispersed, and over-dispersed count distributions. This versatility is achieved by parameterizing the double Poisson distribution using neural networks, distinguishing DDPNs from traditional models based on Poisson or Negative Binomial distributions, which are limited to modeling equal-dispersed and over-dispersed count data, respectively. However, significant weaknesses have been highlighted by the reviewers. These include a lack of rigorous comparative analysis with existing count regression models, a perceived lack of novelty stemming from merely adapting the first layer of a hierarchical model to a double Poisson distribution, and methodological issues with the use of Negative Log Likelihoods (NLLs) to compare models built under different distribution assumptions. Addressing these critical concerns is essential before this paper can be considered suitable for publication.

**Additional Comments On Reviewer Discussion:**

The discussions did not alter the reviewers' initial evaluations, and their opinions remained divergent after the rebuttal. Given the persistence of several valid and critical concerns raised by Reviewers G7zs, xkAG, and NZf8, the AC deems the paper not ready for publication in its current form. These issues must be addressed comprehensively to meet the publication standards.

---

### Decision · Program_Chairs · 2025-01-22

Reject